# Probabilistic greedy algorithm solver using magnetic tunneling junctions for traveling salesman problem

Ran Zhang [1], Xiaohan Li[1], Caihua Wan [1,2,3] ✉, Raik Hoffmann[4], Meike Hindenberg[4], Yingqian Xu[1], Shiqiang Liu[1], Dehao Kong[1], Shilong Xiong[1], Shikun He [5], Alptekin Vardar[4], Qiang Dai[5], Junlu Gong[5], Yihui Sun[5], Zejie Zheng[5], Thomas Kämpfe [4,6] ✉, Guoqiang Yu [1,2,3] & Xiufeng Han [1,2,3] ✉

Combinatorial optimization underpins applications in artificial intelligence, logistics, and network design, yet classical techniques such as greedy search and dynamic programming struggle to balance efficiency and solution quality at scale. We present a probabilistic framework that embeds true random number generators based on spin-transfer-torque magnetic tunnel junctions into a greedy solver. Intrinsic stochastic switching enables configurable random number distributions, which we use to inject controlled randomness via a temperature parameter that interpolates between deterministic and stochastic choices, balancing exploration and exploitation. Applied to the traveling salesman problem, the framework yields high-quality tours and outperforms simulated annealing and genetic algorithms in solution quality and convergence speed. In larger instances with up to 70 cities, it maintains its advantage, reaching near-optimal solutions with fewer iterations and reduced computational cost. These results show that hardware true randomness with tunable statistics can improve heuristic search and motivate integrated, energy-efficient probabilistic hardware for scalable optimization.

Combinatorial optimization is a cornerstone of modern computational science, playing a pivotal role in domains ranging from artificial intelligence and machine learning[1,2] to logistics[3–5] and operations research[6]. The objective is to identify an optimal configuration from a finite but exponentially large set of possibilities, where even modest increases in problem size can render classical methods impractical due to the exponential growth of computational complexity[7]. While deterministic algorithms such as dynamic programming[8] and branch-and-bound[9] have proven effective for small-scale problems, they often fail to scale efficiently to larger scenarios or escape local optima when confronted with the complex landscapes of combinatorial spaces[10].

In recent years, there has been a paradigm shift towards incorporating randomness into optimization algorithms[11–13], leading to a emerging class of techniques termed stochastic or probabilistic optimization[14–20]. Methods such as simulated annealing, genetic algorithms, and Monte Carlo simulations have demonstrated the potential of randomness to diversify search strategies, enabling algorithms to explore solution spaces more comprehensively and escape local minima. However, the efficacy of these methods is highly dependent on the quality and configurability of the random number generators (RNGs) employed[13,21,22]. Traditional RNGs, whether pseudo-random or hardware-based, often lack the flexibility required to dynamically

[1]Beijing National Laboratory for Condensed Matter Physics, Institute of Physics, Chinese Academy of Sciences, Beijing, China. [2]Center of Materials Science and Optoelectronics Engineering, University of Chinese Academy of Sciences, Beijing, China. [3]Songshan Lake Materials Laboratory, Dongguan, Guangdong, China. [4]Fraunhofer IPMS, Center Nanoelectronic Technologies, Dresden, Germany. [5]Zhejiang Hikstor Technology Co. Ltd, Hangzhou, China. [6]TU Braunschweig, Institute for CMOS Design, Braunschweig, Germany. ✉e-mail: wancaihua@iphy.ac.cn; thomas.kaempfe@ipms.fraunhofer.de; xfhan@iphy.ac.cn

adjust their distribution characteristics, limiting their adaptability to different optimization scenarios.

A promising development in this field is the utilization of magnetic tunneling junctions (MTJs) as a source of true random numbers, enabling true random number generators (TRNGs), which exploit inherent physical randomness rather than deterministic algorithmic processes[23,24]. MTJs, typically used in non-volatile memory technologies[25–28], exhibit probabilistic switching behavior that can be finely tuned by external control parameters such as voltage or magnetic field strength. This inherent stochasticity—referred to as hardware randomness due to its direct physical origin—makes MTJ-based TRNGs uniquely suited for probabilistic computing[29–33], where the randomness can be directly mapped onto computational processes. The ability to configure the probability distribution of an MTJ-based TRNG—effectively creating probabilistic bits (p-bits), binary units defined by probabilistic rather than deterministic states—enables an alternative approach to algorithm design, where the degree of randomness can be adjusted in real time to influence decision-making processes[34,35].

Prior conceptual and simulation-level frameworks, such as the SPINBIS spintronics-based Bayesian inference engine built on MTJ stochastic bit-stream generators[36], and spin-orbit-torque-based Bayesian reasoning hardware[37], have demonstrated the feasibility of MTJ-based probabilistic inference in data-fusion and inference tasks. Distinctively, our work goes beyond these precedents by experimentally embedding MTJ-based, probability-distribution-configurable TRNGs into a probabilistic greedy algorithm for TSP optimization. This hybrid hardware-algorithm co-design is, to our knowledge, a representative fully experimental demonstration of Bayesian-PDF-matched probabilistic optimization for combinatorial problems.

In this study, we propose an advanced optimization framework that leverages MTJ-based TRNGs to solve complex combinatorial problems. Specifically, we introduce a probabilistic greedy algorithm for the traveling salesman problem (TSP)[22,38–41] – a canonical example in combinatorial optimization – to showcase the potential of this approach. The TSP challenges a solver to find the shortest possible route that visits a given set of cities and returns to the starting point, and it is well known for its non-deterministic polynomial-hardness (NP-

Hardness). By incorporating MTJ-based TRNGs into the decision-making process, we can modulate the selection strategy for the next city, transitioning smoothly between deterministic greedy choices and purely random selection. This dynamic adaptability enables the algorithm to effectively balance exploration and exploitation, thereby improving its ability to find high-quality solutions efficiently.

## Results and Discussion

Figure 1 presents a detailed characterization of the MTJ-based TRNG employed in this study. The resulting *R-H* hysteresis loops, shown in Fig. 1b, reveals a clear and sharp switching between high and low resistance states, confirming the stability and reproducibility of the MTJ's magnetic switching behavior. The MTJ has a high tunnel magnetoresistance (TMR) ratio ~175%, which is essential for ensuring reliable and distinct resistance states. The MTJ's resistance switching behavior under current pulses is illustrated in Fig. 1c.

By applying a series of current pulses, we observed stochastic switching of the free layer's magnetization, resulting in resistance changes. This stochastic behavior serves as the basis for the MTJ-based TRNG. To further analyze the switching probability, Fig. 1d plots the probability of switching as a function of the applied write voltage. The experimental data (green circles) show a gradual increase in switching probability ($P_{sw}$) with increasing voltage ($V$), which is accurately captured by the fitted sigmoidal curve (black solid line). The fitting parameters $b$ and $c$ indicate the sharpness and offset of the curve.

$$P_{sw} = \frac{1}{1 + e^{-b(V + c)}}$$

This relationship $P_{sw}(V)$ is crucial, as it enables precise control over the probability distribution of generated random numbers. Figure 1e demonstrates the results of continuous resistance measurements at three fixed voltages: 0.275 V, 0.282 V, and 0.288 V, corresponding to $P_{sw}$ of 25%, 48%, and 81%, respectively. These measurements confirm that the device can achieve consistent and repeatable switching behavior, with well-defined probability at each voltage level. The ability to finely tune the switching probability by adjusting the voltage is a key advantage of MTJ-based TRNGs, allowing for the

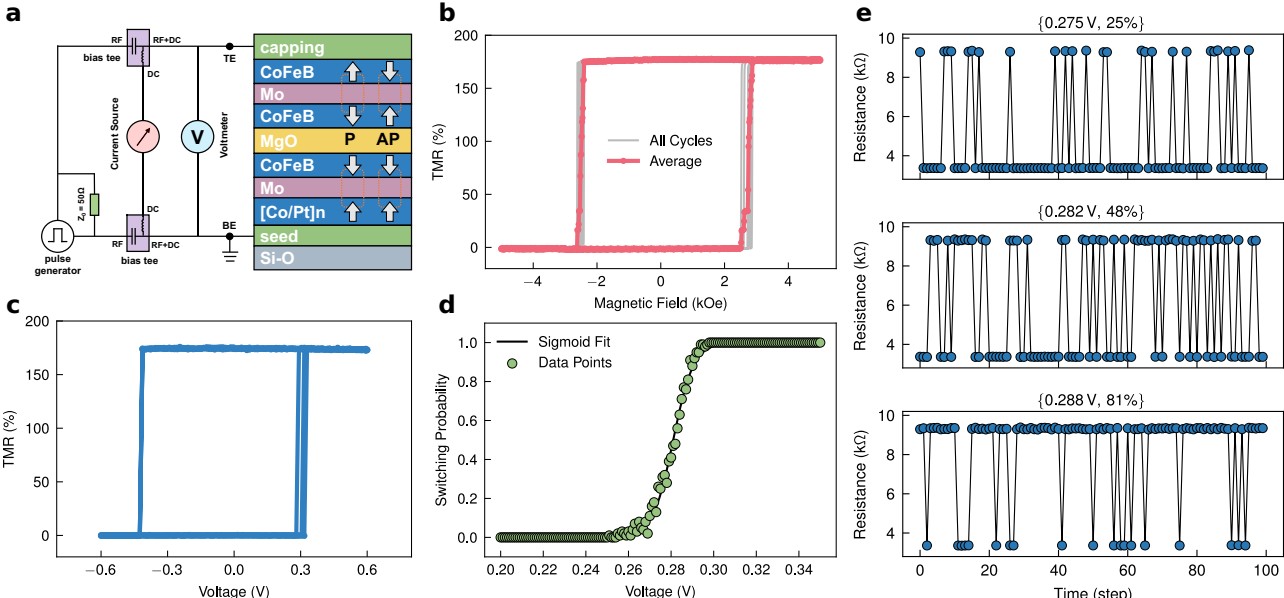

**Fig. 1 | Characterization of the performance of a TRNG based on an MTJ.**
**a** Schematic of the device structure and measurement setup. **b** *R-H* hysteresis loops obtained by sweeping an out-of-plane magnetic field. **c** Resistance switching behavior of the MTJ induced by current pulses, which trigger free layer magnetization switching. **d** MTJ switching probability as a function of applied write voltage. The black solid line represents the fitted sigmoidal curve. **e** Resistance measurements from continuous testing at fixed voltages of 0.275 V, 0.282 V, and 0.288 V, corresponding to switching probabilities of 25%, 48%, and 81%, respectively.

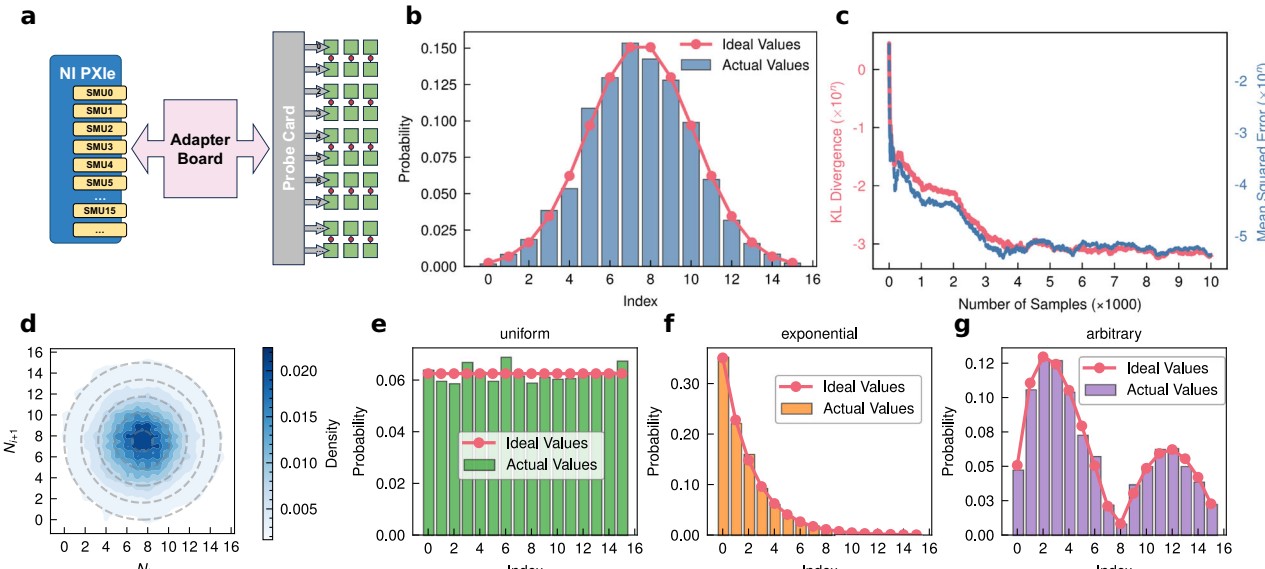

**Fig. 2 | Probability-distribution-configurable TRNGs based on multiple MTJs.**
**a** Schematic diagram illustrating the connection of four MTJs to the NI PXIe system through a probe card and adapter board. **b** Generated random numbers exhibiting a Gaussian distribution. **c** Error analysis of the Gaussian-distributed random numbers, with the left axis representing KL divergence and the right axis representing the mean squared error. **d** Neighbor correlation of the Gaussian-distributed random numbers. The color intensity indicates the sample point density, and the concentric circles indicate weak neighbor correlation. **e–g** Random number generation results for several typical probability distributions: (**e**) uniform distribution, (**f**) exponential decay distribution, and (**g**) user-defined arbitrary distribution.

generation of random numbers with specific statistical properties tailored to various probabilistic algorithms. While achieving fine-grained control over switching probabilities typically requires highly precise voltage tuning, we adopt a hybrid control scheme previously validated in ref. 42, where switching probability is regulated via pulse-width modulation rather than analog amplitude adjustment. This digital control method ensures consistent sigmoidal switching behavior across devices, even when target probabilities are closely spaced. Moreover, the use of a self-stabilizing feedback mechanism compensates for device variation and drift, allowing robust and scalable probability distribution generation without relying on high-resolution voltage sources.

Figure 2 showcases the versatility of the MTJ-based TRNG in generating random numbers with configurable probability distributions. The schematic diagram in Fig. 2a illustrates the experimental setup, where multiple MTJs are connected to the NI PXIe system through a probe card and adapter board (Supplement I). This configuration enables the simultaneous measurement of multiple MTJs, allowing for efficient data collection and parallel testing of different devices.

Figure 2b displays the random numbers generated by the MTJ-based TRNGs. The generated values align closely with the expected Gaussian distribution, as evidenced by the smooth bell-shaped curve. This Gaussian-distributed randomness is achieved by carefully adjusting the write voltage of the MTJs, demonstrating the flexibility of the TRNG in producing specific distributions. The transformation from binary Bernoulli TRNGs into a probability-distribution-function-configurable TRNG modeled as a Bayesian network can be found in Supplement II in details.

To quantitatively evaluate the accuracy of the generated distributions, Fig. 2c presents the error analysis, where the left axis represents the Kullback-Leibler (KL) divergence and the right axis represents the mean squared error (MSE). The KL divergence measures the difference between the experimentally generated distribution and the theoretical Gaussian distribution, while the MSE quantifies the average deviation of the generated values from the expected mean and variance. Both metrics indicate minimal errors, confirming the high fidelity of the MTJ-based TRNG in replicating desired distributions.

The neighbor correlation of the generated random numbers is analyzed in Fig. 2d, where the color intensity represents the sample point density. The nearly uniform distribution of points and the presence of concentric circles indicate negligibly weak neighboring correlation, signifying that the generated random numbers are statistically independent. Our STT-MTJs are not low-barrier ones and each random number is generated by a reset-sampling circle, therefore correlevance between neighboring random numbers no longer an issue here. This property is essential for ensuring that the TRNG can produce high-quality random numbers suitable for applications requiring true randomness, such as probabilistic algorithms and cryptographic operations.

Figure 2e–g demonstrate the capability of the TRNG to generate random numbers following various probability distributions. Figure 2e presents a uniform distribution, where each value has an equal probability of being sampled. Figure 2f shows an exponential decay distribution, characterized by a high probability for smaller values and a rapidly decreasing probability for larger values. Finally, Fig. 2g illustrates a user-defined arbitrary distribution, highlighting the flexibility of the TRNG in generating custom probability profiles. This configurability is critical for integrating the TRNG into a wide range of applications, from stochastic optimization to artificial intelligence, where diverse probability distributions are needed to guide decision-making processes. The probabilistic nature of the algorithm inherently mitigates the influence of occasional transient faults or soft errors in random number generation. Additionally, the embedded self-calibration and stabilization routines further ensure robustness by identifying and correcting persistent anomalous switching behaviors at the hardware level. It is important to note that the experimentally demonstrated capability of generating random numbers with configurable and dynamically tunable probability distributions (Fig. 2) directly supports the probabilistic selection mechanism required by our greedy algorithm in solving the TSP (Fig. 3). At each iterative step of the algorithm, the MTJ-based TRNG efficiently provides random samples precisely matching the dynamically updated probability distribution defined by Eq. (1). This intrinsic alignment between the MTJ-based TRNG capabilities and the probabilistic selection mechanism greatly enhances both the solution quality and computational efficiency, clearly

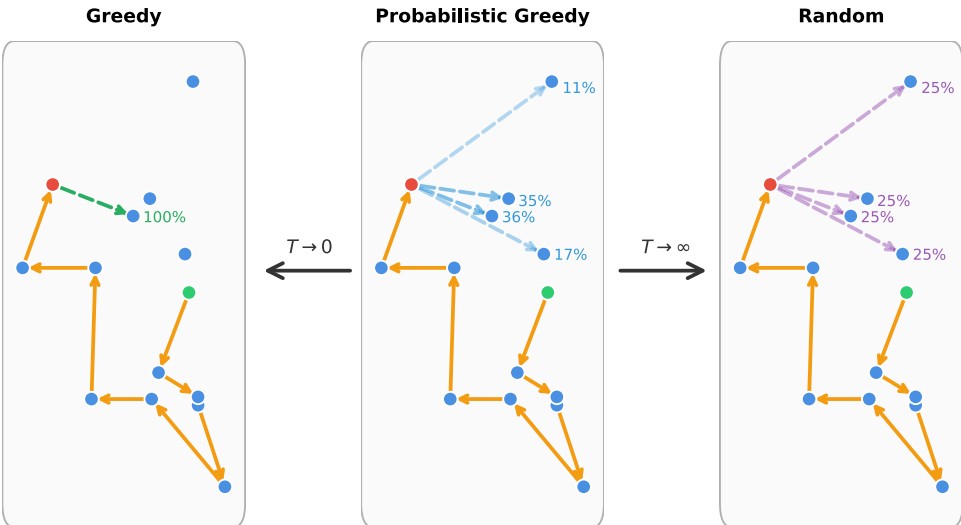

**Fig. 3 | Probability of selecting the next city under different temperature ($k_BT$) conditions.** When $k_BT$ approaches 0, the algorithm always selects the closest city, equivalent to a greedy algorithm. When $k_BT$ approaches infinity, the selection probability for each remaining city is equal, which is analogous to random selection. When $k_BT$ is within a suitable intermediate range, a probabilistic greedy algorithm can be achieved.

distinguishing our approach from traditional algorithms that rely on fixed or less-flexible random number generators.

Maintaining precise switching probability control in large-scale or on-chip implementations is crucial. In our recent work[42], we demonstrated a scalable hybrid control approach in which pulse-width modulation (PWM) replaces the need for high-resolution analog voltage tuning. By adjusting the duration of fixed-amplitude pulses using simple digital logic, the system effectively maintains the desired probability distribution across devices, even under thermal drift and process variation. This strategy significantly reduces the requirement for high-resolution DACs or ADCs, making the architecture well-suited for scalable on-chip integration.

Compared to pseudo-random number generators (PRNGs), MTJ-based TRNGs offer significant advantages in probabilistic computing applications, particularly through their ability to directly generate and dynamically tune probability distributions in real time. Unlike deterministic PRNGs, which typically require additional computational overhead for mapping uniform random outputs into desired distributions, MTJ-based TRNGs inherently produce physically-generated randomness with precisely controllable statistics via simple external parameters (e.g., applied voltage or pulse width). This feature significantly reduces complexity and latency, while also offering enhanced parallelism and power efficiency in hardware implementations.

While this work primarily focuses on distribution configurability, we emphasize that the random bitstreams generated by our MTJs have been rigorously validated[42]. In that study, we conducted comprehensive statistical evaluations using the NIST SP800-22 test suite, confirming that the TRNG outputs exhibit high entropy and pass all standard randomness tests without requiring post-processing. The same device architecture and control protocols were employed in this work, ensuring that the TRNGs used for TSP solving maintain equivalent statistical quality. Additional statistical evaluation of the TRNG-generated bitstreams is provided in Supplement I,I,I. The current bitstream generation rate in our experimental setup is limited by the speed of the NI PXIe data acquisition system, operating at approximately 500 kHz per MTJ. However, the intrinsic switching times of STT-MTJs allow for much faster operation. With high-speed peripheral circuits and optimized on-chip integration, generation rates approaching the GHz range are feasible, as reported in recent high-speed TRNG demonstrations[43]. This positions our MTJ-based TRNGs as

suitable candidates for future high-throughput probabilistic computing applications.

Figure 3 presents the probabilistic greedy algorithm's mechanism for selecting the next city in the traveling salesman problem (TSP) under varying temperature conditions. The algorithm utilizes the MTJ-based TRNG to generate random numbers that influence the city selection process, allowing for a probabilistic adjustment of the greedy strategy. The selection probability $P_{i+1}(\overline{N})$ of the next city $\overline{N}$ is a function of the distance $d_{ij}$ between the current city $N$ and $\overline{N}$ as well as a temperature parameter $k_BT$ as shown in Eq. (1).

$$P_{i+1}(\overline{N}_i) = (1 - b_i)\exp(-d_{N\overline{N}_i}/k_BT)/Z$$
$$Z = \sum_{i=1}^{8}(1 - b_i)\exp(-d_{N\overline{N}_i}/k_BT) \tag{1}$$

Here $b_i$ indicates the accessibility of the $i^{th}$ city and $b_i = 1$ once the $i^{th}$ city has been visited or else $b_i = 0$ if it is to be visited. Thus, the final probability of choosing a specific route $P = \prod P_{i+1}(\overline{N}) \propto \exp[-(\sum d_{ij})/k_BT]$ and, straightforwardly, the shortest route $S = (\sum d_i)_{min}$ has the highest probability to be experimentally sampled. This feature assures the convergence of this probabilistic greedy algorithm. More details can be found in the Supplement V.

It is worth noting that the decision of choosing the next city relies on a probabilistic sampling operation according to the series of probabilities $P_{i+1}(\overline{N})$ with $\overline{N}$ being the city indices to be visited. This probability-distribution-function (PDF) defined by $P_{i+1}(\overline{N})$ changes dynamically step by step, which calls for a random number generator that can output random numbers according to the time-variant PDFs. Fortunately, our TRNGs with configurable PDFs match this requirement well.

When $k_BT$ approaches zero, the algorithm operates as a deterministic greedy algorithm, always selecting the closest city to the current one. In this regime, the probability of choosing the closest city is nearly 100%, leading to rapid but potentially suboptimal solutions due to the algorithm's inability to escape local minima.

Conversely, when $k_BT$ is extremely high, the selection probability for each remaining city becomes nearly uniform, leading to a selection process similar to a random walk. This behavior encourages exploration of the solution space but at the cost of reduced efficiency in converging to high-quality solutions. The optimal performance is

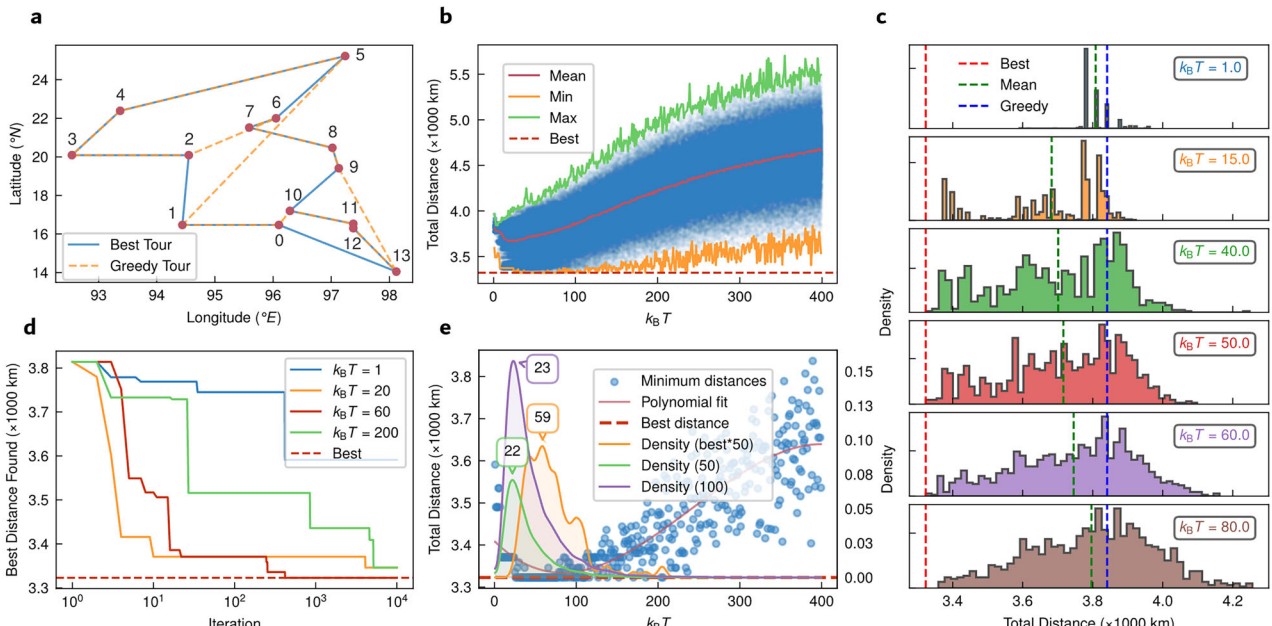

**Fig. 4 | Hardware test results for solving the Burma14 problem using the Probabilistic Greedy Algorithm. a** Map of the Burma14 problem, where the solid line represents the known optimal solution and the dashed line indicates the best solution obtained using the classic greedy algorithm. **b** Total distance statistics of TSP solutions across the range of $k_BT = 1$–400. The red dashed line marks the known optimal solution, while the green, orange and red solid lines connect the maximum, minimum, and average total distances, respectively, obtained at each $k_BT$ value. **c** Distribution of solution distances at six selected $k_B T$ values, showing improved performance and reaching the optimal solution when $k_B T$ is in the range of 40–60. **d** Relationship between the best path distance and the number of iterations for four selected $k_B T$ values. When $k_B T = 60$, the optimal solution can be achieved within 1000 iterations. **e** Scatter plot of the best solutions across $k_B T = 1$–400 (left) and density distribution plots of solutions within 0, 50, and 100 kilometers of the known best solution (right). Each density plot is based on 100 independent solution runs using the probabilistic greedy algorithm.

observed at intermediate $k_BT$-values, where the algorithm effectively balances exploration (randomness) and exploitation (favoring shorter distances), allowing it to escape local optima and discover near-optimal solutions with high probability. Practical considerations regarding frequent adjustments of MTJ-based TRNG distributions have shown minimal overhead. This is primarily because reconfigurations involve only modest voltage adjustments via a small set of parameters. Furthermore, implementing parallel and pipeline operations effectively reduces latency, ensuring these hardware-level adjustments do not significantly affect the overall algorithm performance.

It is important to acknowledge existing epsilon-greedy methods commonly used in reinforcement learning and optimization tasks[44], which similarly balance exploration (randomness) and exploitation (optimal choice). However, our proposed MTJ-based probabilistic greedy algorithm significantly diverges from traditional epsilon-greedy methods in several critical aspects. Unlike epsilon-greedy algorithms, which typically employ a fixed probability ($\varepsilon$) to introduce uniformly random choices, our algorithm continuously and dynamically updates a probability distribution for city selection at every step, based on distances and the adjustable temperature parameter $k_BT$ (Eq. (1)). This dynamic adjustment provides a more nuanced and context-sensitive trade-off between exploration and exploitation. Moreover, the direct hardware-based randomness offered by MTJ-based TRNGs facilitates immediate, real-time, and computationally efficient generation of precisely tuned probability distributions, eliminating the computational overhead associated with transforming uniformly distributed pseudo-random numbers into desired distributions. Consequently, our method achieves superior encoding efficiency, algorithmic flexibility, and scalability, representing a substantial advancement beyond traditional epsilon-greedy approaches.

Figure 4 provides experimental results demonstrating the application of the MTJ-based TRNGs in solving the TSP using the probabilistic greedy algorithm. Figure 4a depicts the map of the Burma14

TSP problem ($n = 14$, where $n$ denotes the problem size), where the solid line indicates the known optimal solution, and the dashed line represents the best solution obtained using a classic greedy algorithm. The probabilistic greedy algorithm, driven by the MTJ-based TRNG, consistently identifies paths that are closer to the optimal solution, as shown by the reduced total distance metrics. Figure 4b illustrates the variation in total distance across a range of $k_BT$ values from 1 to 400. The orange dashed line marks the known optimal solution, while the green, orange, and red solid lines connect the maximum, minimum, and average total distances, respectively, obtained at each $k_BT$ value. The results indicate that the algorithm achieves optimal or near-optimal solutions when $k_BT$ is within the range of 40 to 60, highlighting the significance of selecting an appropriate temperature parameter to balance the probabilistic selection strategy.

Figure 4c further investigates the distribution of solution distances at six selected $k_BT$ values, showing improved performance and reaching the optimal solution when $k_BT$ is between 40 and 60. This analysis underscores the robustness of the probabilistic greedy algorithm in finding high-quality solutions when driven by suitably tuned randomness. Figure 4d examines the relationship between the best path distance and the number of iterations (where one iteration is defined as a single complete solution route, visiting each city exactly once) for four selected $k_BT$ values. When $k_BT = 60$, the optimal solution is achieved within 1000 iterations, demonstrating the efficiency of the algorithm in converging to high-quality solutions. Figure 4e presents a scatter plot of the best solutions obtained across the $k_BT$ range and density distribution plots of solutions within 0, 50 and 100 kilometers of the known optimal solution, further validating the algorithm's effectiveness. For visibility, the density of the 0 kilometer is scaled by a factor of 50. The appearance of a clear peak shape in the distribution plots indicates the existence of an optimal $k_BT$, highlighting the algorithm's sensitivity to temperature parameters in achieving high-quality solutions.

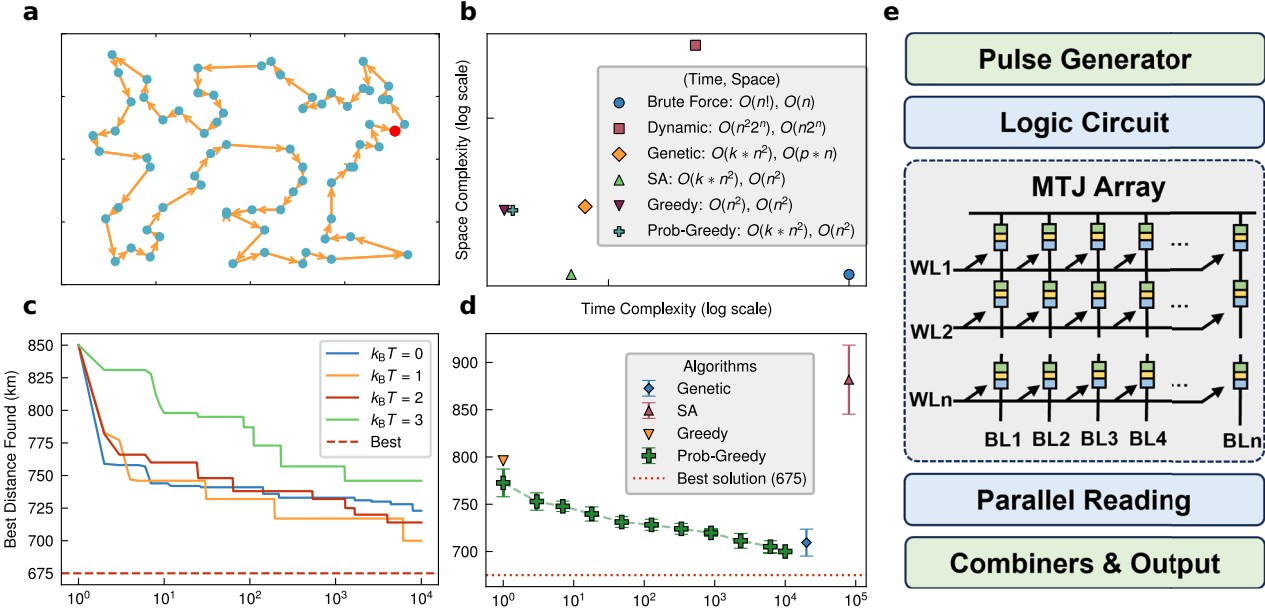

**Fig. 5 | Simulated results with more cities and comparison with other algorithms. a** Map of the st70 problem, where the solid line represents the known optimal solution. **b** A comparison of time complexity and space complexity among Brute Force, Dynamic Programming algorithm, Genetic algorithm, Simulated Annealing (SA) algorithm, Greedy algorithm, and Probabilistic Greedy algorithm, using values for $n = 70$. Axes are unlabeled but proportional. **c** Illustration of how the best path found in the st70 problem decreases as the number of iterations increases for $k_\mathrm{B}T = 0.1, 1.3, 2.0,$ and $3.0$. **d** Comparison of solutions found by heuristic algorithms (Genetic algorithm, SA algorithm, Greedy algorithm, and Probabilistic Greedy algorithm), with the horizontal axis representing the number of iterations. Each data point represents the average of 10 independent runs, with error bars showing standard deviations. **e** A schematic diagram of the core design of a TSP solver based on an MTJ array, highlighting the key components and connections within the solver architecture.

The solver still works well when the city number is increased significantly. Figure 5 illustrates the simulated results obtained with the st70 problem, offering a comprehensive comparison of our algorithm against other established methods. In the map shown for the st70 problem, the optimal solution is indicated by a solid line, serving as a benchmark for evaluating the performance of different algorithms. The st70 problem, with its 70 cities, presents a considerable computational challenge, making it an ideal test case for demonstrating the efficacy of both heuristic and exact algorithms. The comparison of time and space complexity among the algorithms—Brute Force, Dynamic Programming, Genetic Algorithm, Simulated Annealing (SA), Greedy Algorithm, and Probabilistic Greedy Algorithm—clearly highlights the benefits of heuristic methods. While exhaustive approaches like Brute Force and Dynamic Programming struggle with scalability as $n$ increases, heuristic algorithms, particularly the Probabilistic Greedy and Genetic algorithms, strike a balance between computational efficiency and solution quality. This contrast is evident from the results where $n = 70$ is used, showcasing the advantage of these more advanced approaches when tackling larger problems. As the number of iterations grows, the quality of the solutions improves, particularly when varying the thermal fluctuation parameter $k_\mathrm{B}T$. Across different values ($k_\mathrm{B}T = 0.1, 1.3, 2.0,$ and $3.0$), the results indicate that the algorithm's performance is highly sensitive to this parameter, with intermediate values (e.g., 1.3) leading to a more optimal convergence rate. The gradual improvement in path quality with increasing iterations underscores the algorithm's ability to refine its solution over time. When comparing different heuristic approaches—Genetic Algorithm, Simulated Annealing (SA), Greedy Algorithm, and Probabilistic Greedy Algorithm—the results reveal that incorporating stochastic elements, as seen in the Probabilistic Greedy Algorithm, significantly enhances performance. It is worth noting that while the SA can only evaluate the situation of exchanging two cities at a sample, the Probabilistic Greedy Algorithm can take all the remaining cities into account for a single sampling owing to the arbitrary PDF

configurability of our MTJ-TRNGs. Thus, the latter can deal with a higher entanglement degree, which accounts for its faster convergence speed. By avoiding local optima, the probabilistic variant consistently outperforms the classic Greedy Algorithm, especially in later iterations, demonstrating its potential for yielding superior solutions. Moreover, our MTJ-based probabilistic framework can also benefit advanced metaheuristics such as Ant Colony Optimization (ACO) and Particle Swarm Optimization (PSO). By integrating hardware-level randomness, these metaheuristics can systematically enhance their exploration strategies, potentially leading to improved solution quality and faster convergence due to more effective escape from local optima.

We note that the parameter $k_\mathrm{B}T$, which governs the shape of the exponential probability distribution, is currently selected empirically for each problem instance. While effective in practice, developing a more systematic or adaptive strategy for temperature tuning—analogous to annealing schedules or meta-optimization—remains an important avenue for future work, particularly to enhance scalability and generalizability.

Finally, the schematic in Fig. 5e illustrates the core design of a TSP solver based on an MTJ array. This hardware-based solver taps into the inherent randomness of the MTJ array, which can be efficiently reused to generate random numbers of any required length. This modularity and scalability make the MTJ array particularly well-suited for probabilistic algorithms like Simulated Annealing and Probabilistic Greedy Algorithm, enhancing the solver's adaptability for larger and more complex TSP instances. The ability to expand the random number generation capability of the MTJ array according to an arbitrarily customized PDF without sacrificing performance is a crucial innovation, positioning this design as a versatile and efficient solution for hardware-accelerated optimization tasks.

To provide a transparent system-level comparison, we summarize key performance metrics of our prototype platform versus projected FPGA/ASIC implementations in Table 1. The prototype, implemented

**Table 1 | Benchmark comparison between prototype and projected MTJ-based probabilistic TSP solvers**

| Platform | Sampling / Iteration Rate | Time per 14-city solution (~$10^4$ iterations) | Energy per solution (estimated) |
|---|---|---|---|
| Prototype (NI-PXIe + STT-MTJ, ~100 nm) | ~0.5 MHz (instrument-limited) | ~20 ms | ~$10^{-1}$ J (dominated by external instrumentation overhead) |
| Projected FPGA (spin-CMOS co-design) | 10–50 MHz (conservative) | 0.2–1 ms | $10^{-4}$–$10^{-3}$ J |
| Projected ASIC (STT-/SOT-MTJ, 28 nm node) | ≥100 MHz (intrinsic MTJ < 10 ns switching) | ≤0.1 ms | $10^{-6}$–$10^{-5}$ J |

with 100 nm STT-MTJs controlled by NI-PXIe instrumentation, operates at ~0.5 MHz and requires ~20 ms per 14-city TSP solution, with energy dominated by instrumentation overhead. In contrast, projections based on published MTJ switching speeds ( ≤ 10 ns)[45] and intrinsic switching energies (fJ–pJ per event)[35] indicate that a dedicated FPGA or ASIC could achieve sub-millisecond or even sub-0.1 ms solution times with microjoule-level energy consumption. This analysis highlights the large performance headroom available for integrated spintronic probabilistic solvers.

Furthermore, we note that the step-by-step probabilistic decision-making process in our algorithm bears strong resemblance to the autoregressive sampling mechanism employed in large language models (LLMs), where each token is sampled based on a dynamically updated softmax distribution. This conceptual alignment suggests a future direction for integrating MTJ-based probabilistic hardware with AI inference and generation tasks that involve structured randomness.

This paper presents a distinct probabilistic greedy algorithm that utilizes the stochastic properties of MTJ-based TRNGs to solve complex combinatorial optimization problems. By integrating MTJ-based TRNGs with the PDF reconfigurability into the optimization framework, we can dynamically adjust the degree of randomness in the decision-making process, allowing the algorithm to strike an optimal balance between exploration and exploitation. This capability is achieved through the control of a temperature parameter, which modulates the randomness level and enables the algorithm to adapt its strategy based on the problem state.

The effectiveness of the proposed approach is demonstrated through extensive experimentation on the traveling salesman problem. Our results show that the probabilistic greedy algorithm consistently achieves superior performance compared to classical methods such as simulated annealing and genetic algorithms, both in terms of solution quality and convergence speed. When applied to larger problem instances in simulation, the algorithm exhibits excellent scalability and robustness, maintaining a competitive edge even as the number of cities increases to 70. For significantly larger-scale problems involving hundreds or thousands of nodes, practical implementations may require adaptive parameter tuning strategies, parallelization techniques, or decomposition of the problem into manageable subproblems. Nonetheless, there are no fundamental limitations preventing the scalability of our proposed method. The key advantage of this approach lies in its ability to dynamically modulate randomness through the MTJ-based TRNG, which enhances the algorithm's capacity to escape local optima and discover near-optimal solutions efficiently.

In the current implementation, the temperature-like parameter $k_B T$, which governs the exploration-exploitation trade-off, is determined empirically for each problem instance. While this heuristic approach is effective in practice, developing a systematic or adaptive tuning strategy remains an important direction for future work.

Additionally, although we demonstrate hardware results on medium-scale TSP instances (e.g., Burma14), the results for larger problems (e.g., st70) are obtained through algorithm-level simulations. These simulations validate the algorithmic scalability of our approach, while the underlying hardware design—requiring only $\log_2 N$

MTJs for encoding an $N$-choice distribution—offers intrinsic architectural advantages for future large-scale implementations. It is worth noting that, besides the logarithmic scaling of MTJ count ($O(\log_2 N)$), our probabilistic greedy algorithm requires only $O(N)$ auxiliary memory for conditional probability parameters, which remains significantly more favorable than the $O(N^2)$ parameter storage required in Boltzmann or Ising machines.

The integration of MTJ-based TRNGs offers a promising direction for developing hardware-accelerated optimization frameworks, with potential applications extending beyond TSP to other NP-hard problems. This framework can be readily generalized to other combinatorial optimization problems, such as graph coloring or scheduling tasks, by simply redefining the specific cost function and adjusting the temperature parameter accordingly. A concrete adaptation of the probabilistic greedy framework to the graph coloring problem is presented in Supplement VI. Future work will explore the integration of these TRNGs into parallel and distributed computing architectures, as well as their combination with advanced machine learning models to further expand the capabilities of probabilistic optimization methods. This research establishes a solid foundation for leveraging hardware-level stochasticity in computational algorithms, offering additional possibilities for tackling complex optimization challenges with greater efficiency and effectiveness[4].

While the current study emphasizes the feasibility and statistical behavior of MTJ-assisted probabilistic solvers, we acknowledge that absolute runtime, energy, and area efficiency have not been characterized in this work. Our experimental platform involves instrument-level control and is not representative of a fully integrated solution. Future efforts will focus on ASIC- or FPGA-based implementations to enable rigorous evaluation of system-level performance metrics, leveraging the intrinsic speed and low-power characteristics of spintronic devices. Beyond traditional combinatorial problems, our framework also holds potential for accelerating probabilistic AI models, such as autoregressive generators, by serving as a hardware-compatible platform for structured random sampling.

## Methods

The stack structure of the employed STT-MTJ devices[46–48], as depicted in Fig. 1a, is from top to bottom capping/CoFeB/Mo/CoFeB/MgO/CoFeB/Mo/[Co/Pt]$_n$-based synthetic anti-ferromagnetic structure/Seed/SiO$_2$. The multilayer films were deposited by magnetron sputtering on a thermally oxidized silicon substrate under a vacuum environment of $10^{-6}$ Pa. Following deposition, the films were annealed at high temperature in an external magnetic field perpendicular to the film plane. The devices were then patterned into cylindrical STT-MTJs using standard lithography and etching processes. Magneto-transport measurements of the fabricated devices were conducted using an Hprobe H3DM tester. The samples were subsequently connected to a Keysight B1500A semiconductor analyzer and a NI PXIe system through a probe card and adapter board, enabling comprehensive experimental control and data acquisition through a Python-based interface. This setup facilitated precise electrical measurements and switching probability characterization of the STT-MTJs, providing a reliable platform for evaluating their performance as TRNGs. To

ensure stable operation and mitigate the impact of device-to-device variations and long-term drift, we previously developed self-stabilizing techniques and pulse-width modulation strategies for MTJ-based TRNGs. These methods allow each MTJ to autonomously correct its switching probabilities, ensuring consistent random number generation across large-scale device arrays without frequent manual calibration[42,49]. Specifically, for the TSP solver implementation reported herein (e.g., Burma14 problem), four MTJs were utilized to generate the required configurable random distributions. All MTJ devices used exhibited consistent sigmoid-shaped switching probability curves with stable and reproducible behavior, as characterized and verified prior to algorithmic integration.

## Data availability

All data needed to evaluate the conclusions in the paper are present in the paper and available at https://doi.org/10.6084/m9.figshare.28071089.

## Code availability

The code used in this work is available at: https://doi.org/10.5281/zenodo.17503789

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

## Acknowledgements

This work was supported by the National Key Research and Development Program of China (MOST) (Grant No. 2022YFA1402800), the National Natural Science Foundation of China (NSFC) (Grant Nos. 12134017, 51831012, 51620105004, and 12374131), the Strategic Priority Research Program (B) of Chinese Academy of Sciences (CAS) (Grant No. XDB33000000) and the CAS President's International Fellowship Initiative (PIFI) (Grant No. 2025PG0006), awarded to X.H.; C.W. appreciates financial support from the Youth Innovation Promotion Association, CAS (Grant No. 2020008).

## Author contributions

C.W., T.K. and X.H. conceived the research direction, coordinated the collaboration, and supervised all stages of the project. R.Z. carried out the majority of the experiments, including device measurements, data acquisition, and quantitative analysis. R.Z., X.L., C.W., Y.X., S.L., D.K., S.X., G.Y. and A.V. jointly contributed to the development of the experimental concepts and methodology, continuously refined the ideas through discussions, and participated in interpreting the results. R.H. and M.H. supported the characterization workflow and provided technical assistance for the measurement infrastructure. S.H., Q.D., J.G., Y.S. and Z.Z. were responsible for device and sample preparation, including thin-film stack growth, microfabrication, and device processing. All authors contributed to manuscript preparation, provided critical feedback at different stages of writing, and approved the final version of the manuscript.

## Funding

## Competing interests

The authors declare no competing interests.
