## [Transparent Peer Review file · Nature Communications]

Probabilistic Greedy Algorithm Solver Using Magnetic Tunneling Junctions for Traveling Salesman Problem

Corresponding Author: Professor Thomas Kämpfe

Version 0:

Reviewer comments:

Reviewer #1

(Remarks to the Author)

(Remarks on code availability)

Yes, I can install and run the code well.

Reviewer #2

(Remarks to the Author)

The authors of "Probabilistic Greedy Algorithm Solver Using Magnetic Tunneling Junctions for Travelling Salesman Problem" propose a stochastic variant of the traditional greedy algorithm, utilizing the intrinsic randomness of Magnetic Tunneling Junctions (MTJs) to introduce controllable stochasticity. They employ Bayesian inference to dynamically update the Probability Density function (PDF), enabling the mapping of the binary Bernoulli TRNG outputs of MTJs to desired probability distributions. This capability is leveraged in their probabilistic greedy algorithm for decision-making during the optimization process. The authors put forth that this approach reduces complexity while improving solution quality for the NP-hard Travelling Salesman Problem (TSP). The work on using Bayesian inference for dynamic PDF generation with MTJ for TSP is particularly interesting, however, the concept itself is not novel and several aspects require clarification and further verification. Therefore, I cannot recommend it for publication in Nature Communications, and quite substantial improvements are necessary for further consideration.

1. The utilization of MTJs in solving the travelling salesman problem (TSP) is not clearly explained. I recommend that the authors provide a more detailed description. For instance, in Figure 2, the authors present random numbers with various distributions, but the connection between this aspect and the TSP is unclear. The authors should quantify and explicitly illustrate the benefits, making the advantage of this approach more apparent.
2. The authors emphasize the use of TRNGs, but the advantage over commonly used PRNGs remains unclear. PRNGs can effectively generate distributed random numbers, and true randomness is not typically a requirement for probabilistic algorithms like this one, or even for cryptographic operations, as dedicated PRNGs are generally sufficient. The authors should justify the necessity of true randomness in this context and discuss the potential advantages of TRNGs over PRNGs in this specific application.
3. The concept of "iteration" in the manuscript is ambiguous. How is iteration defined in this context? For example, in the case of Burma14, does the iteration count increase by one only when b1 through b14 all become 1, or does it increment with every clock cycle? Clarifying this would enhance the reader's understanding.
4. The probabilistic greedy algorithm proposed by the authors closely resembles the epsilon-greedy search widely used in reinforcement learning and machine learning. Epsilon-greedy methods strike a balance between exploration (randomness) and exploitation (greedy optimization). There is no acknowledgement of these existing methods, as epsilon greedy also combines randomness into the greedy algorithm with a Q learning parameter. There exists literature on these techniques and an instance of TSP implementation as well in "doi.org/10.1049/tje2.12303", the details of which are missing. The authors do not discuss how their approach is unique or the impact of dynamic PDF generation with MTJs.
5. There are specific details on their measurement setup but not on the MTJ devices they are using. Like variation in the

patterned device dimension, the switching curve deviations of MTJs, cycling degradation and the impact of temperature changes on device switching characteristics. Also, what methods are employed to mitigate if such issues are present? The sigmoid switching probability curve should be discussed for all the devices being used to solve the TSP problem and mention the number of devices used to solve the problem.

6. The voltage control as discussed by the authors is highly precise and stabilized with feedback as mentioned in the supplementary. However, there are concerns about how such precision will be maintained when scaling the system to address larger problems. When transitioning to on-chip implementations, maintaining precision will require the integration of high-resolution ADCs and DACs impacting overall system performance.

7. The manuscript lacks a detailed evaluation of the randomness and entropy tests for the TRNG-generated random bitstream data. The authors just briefly touch upon the neighbour correlations of the data generated out of the Bayesian inference. It is also unclear whether the MTJs employed in their setup require postprocessing of the random bitstreams due to device variations, control and environment impacts that alter the device switching characteristics. The speed of random bitstream generation has not been explicitly addressed. Although they mention in their previous work on SOT-MTJ [32] that they pass the randomness tests with post-processing, there is no such discussion in this work.

8. The conversion of binary Bernoulli TRNG to output based on any PDF, especially Gaussian distribution using the Bayesian inference is a versatile approach. The distribution is used as the prior and based on conditional probability the voltages to the MTJ are controlled, generating random numbers in the required probability distribution. The authors outline this already in their previous work in reference [32]. While dynamically computing the PDF for TSP is a good strategy, the conditional probability limits the speed and control of the MTJs as they depend on the outcome of the previous node. The authors need to address and clarify this issue.

9. The voltage precision plays a critical role in achieving the desired probability distributions, and as the problem scales the requirement becomes more stringent. Conditional probabilities that are close in value demand highly precise and stable voltage adjustments, along with a repeatable sigmoidal switching trend. The shape and consistency of the sigmoid curve are crucial for accurately determining the resulting PDF, but variations in device characteristics can significantly impact this curve. There is no discussion on how the authors plan to address such issues when scaling, as they currently use precision instruments to apply the voltage.

10. The authors describe that the kBT value is empirically found for optimal balance between exploration and exploitation. However, this implies that kBT would need to be tuned for every specific problem instance or whenever other hyperparameters of the algorithm are modified. The authors should provide more discussion on how this parameter is affected. One can see that kBT is 60 in the burma14 a 14 city TSP and 1 in the st70 a 70 city TSP suggesting significant variability in its value. Then this search would be similar to the Boltzmann machine-based SA, where each temperature step would govern a local search. Instead of the algorithm finding one on its own, here the optimal temperature is predetermined empirically. Finding this optimal temperature would then become a problem on its own. I suggest that the authors provide a theoretical framework or methodology to determine the optimized kBT for different scenarios.

11. Figure 4(c) and (e) represent the density of solutions, and it is unclear how many solutions form the basis of these density metrics. The metrics can be discussed in terms of the probability of a solution for better clarity. There is no mention of the time taken to solve these TSP instances or any insight into the trend of time to solution as the problem size increases. Only algorithm space and time complexities are provided. The actual time required by their hardware to solve the problem is not discussed. The analysis of the energy and area occupied by their system is also vital, which is absent.

12. The authors have solved st70 TSP by simulating the probabilistic greedy search algorithm. The authors of reference [36] have implemented a similar st70 problem on hardware. With simulation data provided by the authors, a proper comparison cannot be drawn on the performance of the proposed approach. Factors of voltage precision levels or the number of available hardware MTJ devices could restrict larger-scale implementation like the 70-city problem in the probabilistic greedy methods with MTJs. Benchmarking different methods in terms of hardware performance is essential to substantiate the main claim. A scalability analysis should be included, showing the computational costs and performance metrics as the number of cities increases.

13. The information presented in Figure 5 is insufficiently detailed. The subfigures of Fig. 5 should also be cited properly in the main text. In Figure 5b, the axes lack numerical values, and concepts such as time and space complexity are not adequately defined. In Figure 5d, the comparison details are insufficient—are optimal parameters used for the alternative methods? There is only block diagram in Fig. 5e. What will the system performance be expected in Fig. 5e? Some quantitative analysis would be required. While the authors claim superior performance in solution quality and convergence speed compared to classical methods, this is not convincingly demonstrated. I further recommend that the authors scale up the number of cities incrementally using different methods and present the results to better illustrate the performance of their proposed method, and provide a quantitative comparison of their algorithm performance with other state-of-the-art approaches, particularly in terms of solution quality and scalability. Benchmarking against state-of-the-art combinatorial optimization problem (COP) solvers would also help establish the novelty and effectiveness of the proposed algorithm.

Minor comments

1. The paper mentions "n" in the context of scalability but does not define it clearly upon first usage.

2. The statement that "multiple MTJs are connected" lacks specificity regarding the exact number of MTJs.

3. Fig. 1e shows a switching characteristic with respect to time similar to superparamagnetic MTJ, which is against a stable STT-MTJ behaviour. This confusion should be addressed.

4. Measuring only the neighbouring correlation of generated random numbers is insufficient to demonstrate statistical independence or high-quality randomness. The authors should validate the quality of their random numbers by testing the entire sequence using standard statistical tests.

5. Figure 4(b) explanation on Page 8 of the main text has an error in identifying the curves.

6. In Figure 4(e) what is Density (best*50), is it a distribution plot of solutions with 0 kilometers of the known optimal as mentioned on Page 8 of the main text? And the numbers on the peak are the maximum number of solutions giving exact distances? More clarity is required.

7. Figure 5(d) shows the results of different heuristic algorithms with the error bars, there is no mention of how many samples is the error bar drawn. Prob-Greedy data is not present for 10^5 iterations in solving st70 on simulation?
8. In the description and main text explanation of Figure 4(d) optimal solution with 1000 repetitions is used, but in the graph the axis says iterations. Are repetitions and iterations used interchangeably? Or are the authors performing 1000 repetitions of the iterations?
9. More quantitative description and analysis would be beneficial in Supplemental Information.

(Remarks on code availability)

Reviewer #3

(Remarks to the Author)

The manuscript introduces a probabilistic greedy algorithm based on stochastic Magnetic Tunnel Junction (MTJ) in the framework of probabilistic computing. Through dynamically modulating the degree of randomness, the algorithm herein has manifested the performance in combinatorial optimization problems. The authors claim that, when applied to the Traveling Salesman Problem, this algorithm surpasses other classical approaches, characterized by the solution quality and more rapid convergence speed.

Overall Evaluation: This work is overall coherent. However, it does not meet the high standard of Nature Communications due to the lack of both novelty and enough new results:

(1) The key experimental finding in this work is that four asynchronous MTJs are utilized to generate the desired probability distribution function (Figure 2). These results have already been reported and well-studied in a previous paper by the same group [Advanced Science, 11(23) 2402182 (2024)]. The probability distributions are the same, just with a different set of measurement data.

(2) The probabilistic greedy algorithm highlighted in this work is also not new. The concepts and algorithms have been previously reported in several papers many years ago [Computers & Operations Research, 37: 432 (2010); IEEE Transactions on Communications, 58: 3286 (2010); Proceedings of the 42nd IEEE International Symposium on Multiple-Valued Logic (ISMVL), Victoria, CANADA, F 2012 May 14-16, (2012)].

(3) The scale of the problem solved (14 cities by experiment and 70 cities by simulation) is not large enough as compared to a recent experimental work [ref. 36 published in NC] where 80 sMTJs are used in hardware to solve the 70-city TSP.

Considering these facts, this work only has incremental contribution compared to the established literatures. It is suitable to publish in a more specified journal with a few technical issues properly addressed.

Technical Concerns:

- (1) The manuscript mentions that the magnetization direction of the free layer of the MTJ is altered by applying current pulses. However, it fails to specify the magnitude and duration of the applied pulses, as well as the measurement rate. This is important when comparing different algorithms in various platforms (CPU, FPGA etc.).
- (2) Four MTJs are utilized to generate the desired probability distribution function through a Bayesian network. Nevertheless, the impact of the differences among MTJs on the probability distribution function is not discussed in the manuscript. It is well-known that device-to-device variations have deep impact on the performance of p-computers.
- (3) The results presented in Figure 4c indicate that the average solution still has a notable deviation from the correct solution. An in-depth analysis of the underlying causes should be provided. Moreover, there is a discrepancy between the colors of the curves depicted in Figure 4b and the corresponding descriptions in the manuscript, which may cause confusion for readers.
- (4) The simulation in Figure 5c indicates that the algorithm's performance is more sensitive to the parameter $k_B T$. Then, for problems of a larger scale, will there be a situation where the differences in $k_B T$ are extremely small and indistinguishable?
- (5) Although this algorithm outperforms the traditional greedy algorithm in solving the Traveling Salesman Problem (TSP), whether it is equally effective for other types of combinatorial optimization problems remains unclear.
- (6) In Figure 1d, the data points are in green, but they are miswritten as "blue circles" in the manuscript.

(Remarks on code availability)

Version 1:

Reviewer comments:

Reviewer #1

(Remarks to the Author)

Authors have addressed all of my concerns, I recommend to publish as it is.

(Remarks on code availability)

Reviewer #2

(Remarks to the Author)

The authors made substantial modifications to the original manuscript. The revised manuscript presents an engineering link-up between an MTJ-based, PDF-configurable TRNG and a probabilistic-greedy TSP solver. Unfortunately, both ingredients are documented in the literature, and the present version does not yet provide the data or analysis needed to elevate the work from incremental to transformative. Some critical aspects are still missing. Thus, I recommend the authors revise their manuscript according to the comments below.

The authors state that their hardware scales with $\log_2 N$, but I find this argument unconvincing. In practice, the dominant factor in scaling is memory usage. For instance, solving an N-city TSP typically requires storing N sets of information, implying at least $O(N)$ memory. If we focus solely on the use of MTJs, algorithms such as simulated annealing can reuse a single MTJ across problem sizes without any sacrifices in time or energy. However, this does not mean that the hardware is free from scaling considerations, while the MTJ count may remain constant, other aspects, particularly memory, still scale with problem size.

The rebuttal states that the new MTJ system over [Adv. Sci. 11 (23) 2402182 (2024)] offers “significantly higher SNR, stability and control resolution.” while no evidence is provided. Authors need to present a single figure or table that overlays old-versus-new statistics, such as KL-divergence to target PDFs, NIST SP 800-22 pass counts, and BER histograms. Without these metrics, the improvement (and its necessity for TSP quality) remains unsubstantiated.

The authors need to position the work correctly against prior probabilistic solvers. The manuscript currently asserts “no prior integration of Bayesian PDF with MTJ randomness” yet simulation or conceptual precedents exist (e.g. SPINBIS Bayesian-inference engine). Authors need to revise the Introduction to acknowledge these studies explicitly, and then specify what is new here.

The authors need to provide transparent device and system benchmarks inside this paper. Key specs are scattered across earlier publications; readers cannot verify or compare. The authors need to add a table providing prototype time-to-solution and energy (NI-PX1e loop) plus projected ASIC/FPGA numbers based on published MTJ switching speeds.

I suggest adding Figures R2 and R5, along with their corresponding discussions, to the Supplementary Information, as they enhance understanding.

Figure 4(b) explanation on Page 8 of the main text has an error in identifying the curves.

There is no logarithmic scaling in Fig. 5b. Please have a check “As such, precise numerical values are not included on the axes, but the figure is drawn proportionally to reflect known theoretical trends (e.g., logarithmic vs. linear vs. exponential scaling).”.

The authors have resolved the issue in the figure caption, but the mismatch remains in the main text. The authors can make the changes accordingly.

(Remarks on code availability)

Reviewer #3

(Remarks to the Author)

The authors have made several clarifications over the previous version. Regarding the novelty and significance concerns, the authors suggest an improved sMTJs quality and PDFs owing to better experimental setup, and compared the difference of greedy algorithms used here with previous literatures.

After assessment, I think the only distinct advantage of this work is the reduction on the number of p-bits used (from N^2 to $\log_2 N$) in the TSP problem studied. However, whether such p-bit scaling advantage can be well-applied to other COP problems is not demonstrated. The author stated in the rebuttal letter that “we have added a detailed adaptation example for graph coloring in the Supplementary Information”, but I'm not able to find any information and evidence on this matter. The supplementary file appears to be the same as the previous version. Overall, I still think this work is better suited for a more specialized journal.

(Remarks on code availability)

Version 2:

Reviewer comments:

Reviewer #2

(Remarks to the Author)

The authors have addressed the issue I previously flagged, and it can now be considered for publication.

(Remarks on code availability)

Reviewer #3

(Remarks to the Author)

The authors have renewed the supplementary information and demonstrated the potential scalability over a graph coloring example which addressed my concern.

(Remarks on code availability)

Response Letter to Reviewers - NCOMMS-24-82665

We sincerely appreciate the reviewers for their time and effort in carefully evaluating our manuscript and for providing insightful and constructive feedback. We have carefully considered all comments and incorporated our responses into the revised manuscript as appropriate. We believe that our revisions effectively address the reviewers' concerns. Below, we provide a point-by-point response to each of the reviewers' comments.

Reviewer #1

Comment: The manuscript, entitled “*Probabilistic Greedy Algorithm Solver Using Magnetic Tunneling Junctions for Traveling Salesman Problem,*” presents a forward-looking approach to hardware-accelerated combinatorial optimization. By leveraging the inherent stochastic behavior of spin-transfer torque magnetic tunneling junctions (STT-MTJs), the authors demonstrate a probabilistic greedy algorithm capable of addressing the traveling salesman problem (TSP) with notable efficiency. Through the adjustable true random number generation of MTJs, the work enables dynamic probability distribution configurations and achieves a smooth transition between purely greedy and fully random search strategies by tuning a “temperature” parameter.

Response: We sincerely appreciate the reviewer’s thoughtful assessment of our work and their recognition of its novelty in the context of hardware-accelerated combinatorial optimization. Our approach aims to harness the intrinsic stochasticity of STT-MTJs to introduce a tunable probabilistic greedy algorithm, and we are pleased that the reviewer acknowledges both the fundamental innovation and the practical implications of this design. The ability to dynamically configure any arbitrarily defined probability distribution as required through MTJ-based TRNGs enables a smooth and controlled transition between purely greedy and fully random search strategies, which we believe represents a valuable advancement in probabilistic computing. We are encouraged by the reviewer’s positive evaluation and are grateful for their insightful perspective on the potential impact of our methodology.

Comment: Beyond its methodological novelty, this work is substantiated by thorough experimental results and simulations. The demonstrations on both a smaller-scale TSP (Burma14) and a larger instance (st70) highlight the scalability and robustness of the proposed solver. In doing so, the

authors provide compelling evidence that integrating configurable MTJ-based TRNGs can significantly improve solution quality and convergence speed when compared to classical approaches such as simulated annealing and genetic algorithms. Especially, they demonstrated that the encoding efficiency using these PDF-configurable TRNGs ($\log_2 N$) is much higher than that of the classic Boltzmann machines for the TSP problems (N^2). Furthermore, the combination of hardware-level randomness with a carefully designed probabilistic search algorithm appears to be a promising direction for tackling NP-hard problems. The authors' success with TSP underscores the potential for broader applications in fields such as scheduling, graph partitioning, and other complex optimization challenges.

Response: We greatly appreciate the reviewer's recognition of the robustness and scalability of our approach, as well as the comprehensive experimental validation that supports our findings. Demonstrating the effectiveness of our solver across different TSP instances, from small-scale (Burma14) to larger-scale (st70), was an essential aspect of our study, and we are pleased and grateful that the reviewer found these results compelling. In particular, the acknowledgment of our encoding efficiency advantage over classical Boltzmann machines ($\log_2 N$ vs. N^2) is highly encouraging and insightful, as it highlights the fundamental efficiency gains enabled by our newly developed PDF-configurable TRNGs compared to the classical Ising or Boltzmann machines. Additionally, we appreciate the reviewer's perspective on the broader applicability of our approach beyond TSP, including potential extensions to scheduling, graph partitioning, and other NP-hard problems. We have made refinements to the manuscript to clarify these aspects and further emphasize the generalizability of our probabilistic framework.

Comment: Overall, this manuscript provides a compelling contribution to the burgeoning field of probabilistic computing. The implementation of MTJ-based TRNGs in a customizable way — together with a carefully orchestrated algorithmic structure — illustrates a path toward higher-performance solutions for large-scale optimization tasks. The multi-faceted evaluation, spanning device fabrication, experimental verification, and algorithmic benchmarking, is particularly commendable. In my opinion, the paper is well-structured, clearly written, and of direct interest to a broad cross-section of researchers working on spintronics, computational optimization, and hardware-accelerated algorithms. I therefore **strongly recommend** its publication in *Nature*

Communications, provided the authors address the questions and minor enhancements noted below. In addition, I suggest adding further references that connect this work to foundational methods in combinatorial optimization and TRNG-based hardware design. I believe clarifications on these matters, alongside the inclusion of key additional citations, would further strengthen the manuscript's narrative and ensure maximum impact for a wide readership.

Response: We are sincerely grateful for the reviewer's strong endorsement of our manuscript and their recognition of its relevance across multiple research domains, including spintronics, computational optimization, and hardware-accelerated algorithms. The reviewer's positive remarks on the structure and clarity of the manuscript are especially valuable, as we have strived to present our findings in a rigorous yet accessible manner. In response to the reviewer's suggestion, we have carefully reviewed and expanded our references to better connect our work to foundational research in combinatorial optimization and TRNG-based hardware design. These additional citations provide further context and reinforce the contribution of our approach within the broader landscape of probabilistic computing. Worth noting, currently, we have also found that our PDF-configurable TRNGs with the structure of a Bayesian Inference Network have shared exactly the same network architecture as the autoregression model widely adopted in many Large Language Models (LLMs). In this autoregression model, the probability of sampling a new token depends on the conditional probabilities of previously involved tokens, which is actually similar to what we have done in the probabilistic Greedy algorithm, where tokens are replaced by cities. This similarity also highlights the applicability of our developed methodology in generative AI, particularly in LLMs.

Following this general response, we provide a detailed, point-by-point reply to each of the reviewer's specific comments and questions. We appreciate the reviewer's constructive feedback and have made careful revisions to ensure the manuscript is as clear, comprehensive, and impactful as possible.

Question #1-1: The authors provide strong experimental evidence of repeatable MTJ switching probabilities at varying voltages. However, can the authors expand on how larger-scale device variations — such as minor manufacturing inconsistencies or drift over time — might affect solution quality? Are there calibration strategies to ensure consistent TRNG behavior across an array of MTJs?

Response: We appreciate the reviewer’s insightful comment on the impact of large-scale device variations, including minor manufacturing inconsistencies and long-term drift. These challenges, objectively existent and practical for current state-of-the-art technologies^{1,2} as realized by the referee, have been indeed carefully considered in our recent work, where we proposed strategies to ensure stable operation of MTJ-based TRNGs across large device arrays. In one of our published papers³, we introduced a self-stabilizing approach that dynamically adjusts the sampling probability based on an MTJ’s current state. This methodology enables devices to automatically and quickly converge to the desired switching probability over time without pre-calibration, effectively mitigating initial state differences and slight fabrication variations.

Building on this, our recent work⁴ further addresses drift by exploring alternative control strategies. While voltage adjustment is a conventional method for correcting deviations, fine-tuning voltage amplitude with high precision can be impractical in engineering applications. Instead, we propose pulse width modulation (PWM) as a more feasible and precise approach, as adjusting pulse duration is easier to implement while still significantly influencing switching behavior. By embedding an on-chip calibration feedback loop that periodically evaluates each MTJ’s characteristics, our system can dynamically maintain the intended probability distribution, ensuring reliable random number generation despite device-to-device variations or environmental fluctuations. Overall, adopting pulse width modulation as a practical engineering solution enhances the robustness of the TRNG, making it well-suited for applications requiring high-quality randomness, such as secure communications and probabilistic computing.

In the revised manuscript, we added some discussion on the above point at the end of Methods:

“To ensure stable operation and mitigate the impact of device-to-device variations and long-term drift, we previously developed self-stabilizing techniques and pulse-width modulation strategies for MTJ-based TRNGs. These methods allow each MTJ to autonomously correct its switching probabilities, ensuring consistent random number generation across large-scale device arrays without frequent manual calibration.”

Figure R1. The Downcalibration-2 strategy outputs target switching probabilities of 20% (a) and 80% (b). The top panels show the resistance switching behavior over time, the middle panels present the dynamically adjusted driving voltage of the MTJ during each calibration step, and the bottom panels illustrate the rolling probability (calculated over the last 50 steps). In both cases, the system starts with randomly initialized voltage conditions, rapidly and automatically converges to the target probabilities, and stabilizes with high precision⁴.

Question #1-2: The manuscript focuses on the traveling salesman problem as a case study. Could the authors elaborate on how seamlessly their approach can be adapted to other combinatorial problems, for example, graph coloring or scheduling? Would the same “temperature” tuning principle be sufficient, or might additional modifications be required?

Response: Thank you for raising this excellent point. We focused on the TSP as a representative NP-hard problem to demonstrate the efficacy of our framework, but the underlying principles are indeed applicable to a wide range of combinatorial optimization tasks. The key lies in formulating each problem’s Loss (or Cost) function and incorporating it into the probabilistic decision-making process. For example, in graph coloring, the algorithm would probabilistically select how to color each node to minimize the number of conflicts, while in scheduling, it would seek to reduce overall makespan or resource contention. The “temperature” parameter would again play a similar role in balancing exploration and exploitation: at low temperature, the algorithm favors local minima; at high temperature, it explores more globally. Although the specific probability distribution shaping may need fine-tuning to reflect the constraints or cost landscape of each problem, the overarching method — leveraging MTJ-generated randomness within a probabilistic greedy framework —

remains largely intact. Therefore, we anticipate that only minor modifications, such as redefining the cost function and adjusting the temperature schedule, would be required and enough to generalize our approach to other NP-hard scenarios.

Here we can easily adopt the idea of probabilistic greedy algorithm into the graph coloring problem with the basic equations as follows. The graph coloring problem requires assigning colors to the vertices of a graph $G(V, E)$, such that no two adjacent vertices share the same color, while minimizing the total number of colors used. To adapt our MTJ-assisted probabilistic greedy framework, we define a probabilistic selection process for color assignment at each step, guided by a dynamically adjusted probability distribution based on a cost function. Let us define:

- $V = \{v_1, v_2, \dots, v_n\}$: the set of vertices,
- $C = \{c_1, c_2, \dots, c_k\}$: the set of available colors (with k large enough to allow a feasible coloring),
- $\text{conflict}(v_i, c_j)$: the number of adjacent vertices of v_i already assigned color c_j .

For each vertex v_i , we define the cost of assigning color c_j as:

$$E_{ij} = \lambda \cdot \text{conflict}(v_i, c_j)$$

where $\lambda > 0$ is a penalty weight that amplifies the cost of conflicts.

The probability of assigning color c_j to vertex v_i is then determined by the Boltzmann distribution:

$$P_{ij} = \frac{e^{-E_{ij}/(k_B T)}}{\sum_{l=1}^k e^{-E_{il}/(k_B T)}}$$

This formulation ensures that color assignments with fewer conflicts are more probable, especially at lower temperature T , while allowing more exploration at higher T .

The coloring process proceeds as follows:

1. Initialize all vertex colors to unassigned.
2. For each uncolored vertex v_i , compute P_{ij} for all available colors $c_j \in C$ based on the current partial coloring.

3. Use MTJ-based TRNG and PDF control to sample from P_{ij} and assign color c_j to v_i .
4. Repeat steps 2 – 3 until all vertices are colored.

Optionally, the total number of used colors can be minimized by starting from a conservative upper bound k , and gradually reducing it while checking feasibility.

This probabilistic greedy strategy preserves the same core elements from our TSP implementation:

- The stochastic nature is governed by tunable $k_B T$,
- MTJ-generated Bernoulli bits are used to construct arbitrary PDFs for decision-making,
- The decision process is localized and sequential, making it naturally scalable.

This adaptation highlights the generality of our method: by redefining the cost function E and updating the probability calculation, the same hardware-friendly framework can be extended to a wide class of combinatorial optimization problems.

Question #1-3: The authors note that the probability distribution used in each city-selection step changes over time. Is there any significant overhead associated with frequently reconfiguring the MTJ TRNG voltage settings or readout circuitry during an iterative process? A brief discussion of potential timing constraints or hardware-level overhead would further clarify the performance.

Response: We appreciate this question, as practical considerations are crucial when integrating hardware-level randomness into iterative algorithms. Our experiments indicate that the overhead of updating the MTJ-based TRNG's probability distribution is relatively modest, mainly because the voltage reconfiguration is achieved through adjusting a small set of control parameters. In practice, each step involves updating the “temperature” or weight parameters of the Bayesian network, which can be done efficiently via well-defined write signals without requiring a complete reset of the entire system. Additionally, parallelism helps mitigate reconfiguration delays: while one set of MTJs is being tuned or read, another set can continue generating random numbers according to a previously established distribution. This pipeline approach reduces idle time and keeps the overall process moving at an acceptable rate. As a result, the timing constraints introduced by dynamic PDF reconfiguration have not posed a significant bottleneck in our tests. Of course, if one were to scale the system to very large arrays of MTJs, careful hardware design and scheduling would become

essential to keep reconfiguration overhead low, but we remain optimistic that such optimizations can be handled without drastically affecting the algorithm’s runtime performance, considering the bottleneck of probabilistic computing generally lies in the high-quality prediction of PDF towards convergence instead of PDF reconfiguration; the former has been ensured by the Boltzmann distribution and the corresponding annealing strategy in our probabilistic greedy algorithm.

Question #1-4: While the authors compare their method to simulated annealing and genetic algorithms, have they considered advanced metaheuristics like ant colony optimization or particle swarm optimization? Adding a sentence or two on how the proposed solver might be positioned relative to these approaches could strengthen the discussion.

Response: We appreciate the reviewer’s suggestion to extend our discussion to advanced metaheuristics such as ant colony optimization (ACO) and particle swarm optimization (PSO). In the revised version of the manuscript, we will include a dedicated paragraph to position our proposed solver relative to these approaches. More specifically:

1. Integration of Hardware-Level Randomness

Both ACO and PSO rely on iterative improvement processes guided by decentralized, swarm-like interactions. In ACO, artificial ants traverse a problem space by depositing and following pheromone trails, whereas PSO tracks the dynamic “velocity” and “position” of each particle as it searches for an optimal or near-optimal solution. A natural opportunity arises to inject the hardware-level randomness from our STT-MTJ-based TRNGs into these updates — whether it can introduce controlled noise in the pheromone intensities or generating stochastic velocity perturbations for each particle. By exploiting dynamically reconfigurable probability distributions, researchers could experiment with new methods of diversifying search trajectories and accelerating the escape from local optima.

2. Enhanced Exploration vs. Exploitation

A common theme in metaheuristics is balancing exploration (broad search) and exploitation (refining promising areas). In ACO, for example, highly valued paths become strongly reinforced with pheromones, potentially causing premature convergence if the algorithm lacks sufficient randomness to explore alternatives. Our MTJ-based TRNG framework can

help mitigate such issues by regulating the level of probabilistic deviations during the ant's decision process, thereby preserving a baseline of exploration. PSO encounters a similar trade-off, where particles risk clustering around a suboptimal region; injecting hardware-level randomness in velocity updates could systematically introduce controlled exploration at each iteration, comparable to adjusting “inertia weight” or “constriction coefficients” in standard PSO formulations.

3. Potential Benefits for Convergence

Another compelling aspect is speed. One challenge in advanced metaheuristics is the computational overhead needed to manage multiple agents, track their interactions, and update parameters. By using MTJ arrays as direct sources of random draws, we sidestep purely software-based or pseudo-random routines, potentially streamlining the stochastic elements that underpin ACO or PSO. This integration might lead to faster or more energy-efficient convergence, particularly if the TRNG can be implemented in parallel, allowing different “ants” or “particles” to sample random numbers simultaneously with minimal latency.

4. Implications for Broader Problem Classes

Although we center our discussion on TSP, ACO and PSO are routinely applied to a broad range of NP-hard problems (e.g., scheduling, constraint satisfaction, continuous optimization). Because the probabilistic framework we propose does not depend on problem-specific features but rather on a generalized cost function and temperature-like control, we anticipate that the underlying methodology would translate well into these contexts. Adjustments would primarily involve mapping the problem variables (e.g., partial schedules or partial graph colorings) onto the corresponding algorithm steps (pheromone reinforcement, velocity update, etc.) and then injecting MTJ-based randomness wherever it is most beneficial for diversified exploration.

In sum, we view our probabilistic hardware solver as highly complementary to advanced metaheuristics: it provides a physically grounded, tunable source of randomness that can be tailored to the needs of algorithms like ACO and PSO. By adding a few sentences in the revised text, we

aim to highlight these synergies more explicitly, illustrating how our MTJ-based TRNG could be a valuable resource not just for probabilistic greedy approaches, but also for a wider family of techniques that seek efficiency and resilience in large-scale optimization.

In the revised manuscript, we added some discussion on the above point as follows:

“Moreover, our MTJ-based probabilistic framework can also benefit advanced metaheuristics such as Ant Colony Optimization (ACO) and Particle Swarm Optimization (PSO). By integrating hardware-level randomness, these metaheuristics can systematically enhance their exploration strategies, potentially leading to improved solution quality and faster convergence due to more effective escape from local optima.”

Question #1-5: While the manuscript discusses results for up to 70 cities in TSP, many real-world applications involve hundreds or even thousands of nodes (e.g., in large logistics networks). Do the authors anticipate any significant performance bottlenecks or algorithmic adjustments needed to maintain solution quality or convergence speed at much larger scales?

Response: We appreciate this insightful question regarding the applicability of our approach to more extensive TSP scenarios and, by extension, other large-scale optimization tasks. While our work currently demonstrates success up to 70 cities, we recognize that genuine industrial or logistical problems often involve hundreds or thousands of nodes. From an algorithmic perspective, the core probabilistic greedy principle should, in principle, remain effective at larger scales because the temperature or simulated annealing mechanism (or an equivalent parameter) can still balance exploration and exploitation across a broader decision space. However, as the number of nodes increases, two primary considerations arise:

1. **Time Complexity:** Even though we have shown gains over classical methods, any TSP solver contends with inherent combinatorial growth. Our hardware approach helps mitigate this by accelerating the randomness generation and selection processes, but we would still expect solution times to increase with larger node counts. We are therefore exploring strategies such as parallelizing or “tiling” portions of the problem across multiple MTJ arrays to handle partial sets of cities in tandem, then merging these subproblem solutions, the so-called divide-and-conquer strategy as used in Ref.⁷ or Ref. [36] in the main text.

2. **Convergence and Parameter Tuning:** With a larger search space, maintaining an appropriate balance between exploitation and exploration becomes more delicate. We foresee a need for adaptive temperature schedules or multi-phase temperature schemes that initially promote greater exploration (to avoid early local optima) and gradually shift to more exploitative modes. While the existing mechanism is already flexible, large-scale instances may benefit from “meta-heuristic” tuning or automatic temperature adaptation.

Overall, we do not see any fundamental barrier to scaling up, but achieving comparable solution quality and convergence speed will hinge on judicious parameter tuning, possible hardware parallelism, and potentially breaking down massive TSPs into more manageable subproblems.

Question #1-6: In practical systems, transient faults or soft errors (e.g., due to electromagnetic interference) can occasionally produce outlier random values. Do the authors foresee the need for error-correction logic in the random number generation process, or is the probabilistic nature of the algorithm itself sufficient to absorb occasional anomalies?

Response: We also appreciate the query concerning transient faults or soft errors, which can indeed occur in real-world electronics due to electromagnetic interference, cosmic rays, or other environmental factors. Within our hardware-based stochastic framework, occasional random outliers are less likely to derail performance for two reasons:

1. **Probabilistic Nature of the Algorithm:** Since our approach inherently relies on sampling from a probability distribution, small deviations or occasional spurious values tend to be absorbed in the normal iteration-to-iteration variability of the algorithm. A single outlier in a random sample is less detrimental here than it would be in deterministic logic, especially when repeated samplings guide the search over many iterations. We can actually treat those transient faults or soft errors, if any, as random initializations or resets of networks, which is exactly what we have done in a long-term optimization process.
2. **Built-in Redundancy and Calibration:** In addition, any self-calibration or self-stabilizing measures we employ for device variation can also help detect and correct persistent anomalies in MTJ switching statistics. If a particular MTJ or set of MTJs starts exhibiting

highly aberrant behavior, the calibration routine would detect deviations from the expected distribution and adjust voltage levels or exclude the faulty device(s) from the random pool.

That said, if a system were deployed in highly critical or noisy environments, more explicit error-correction or majority-voting schemes at the TRNG stage could be considered. Such measures would provide an extra layer of confidence. However, for most use cases and standard operating conditions, we have found the inherent resilience of the probabilistic approach sufficient to mitigate issues arising from occasional soft errors.

Question #1-7: The temperature-related parameter ($k_B T$) appears in several figures and equations. A single consistent notation (e.g., always using $k_B T$ or T) throughout the text, figures, and captions would help readability.

Response: We thank the reviewer for highlighting this important point regarding notation consistency. In response to this comment, we have carefully revised the manuscript and figures to uniformly adopt the notation $k_B T$ throughout the entire paper, including equations, figure captions, and main text descriptions. This modification significantly enhances readability and ensures a clear, consistent representation of the temperature-related parameter across all sections.

Question #1-8: The paper occasionally shifts between “true random number generators,” “hardware randomness,” and “probabilistic bits.” Standardizing the terminology — perhaps defining each concept at the outset — would streamline the reader’s experience.

Response: We sincerely appreciate the reviewer’s helpful suggestion regarding the terminology standardization. To enhance readability without disrupting the manuscript flow, we have subtly integrated concise explanations of the terms “true random number generators (TRNGs)”, “hardware randomness”, and “probabilistic bits (p -bits)” within their first mentions in the Introduction section. This approach naturally clarifies the concepts while maintaining the manuscript’s coherent narrative style.

Question #1-9: In addition to the classical references to combinatorial optimization and TSP, it might be worthwhile to cite a few seminal articles on spintronic TRNGs or in-memory computing approaches. This could underscore how the present work builds on, and differs from, earlier device-centric studies.

Response: We thank the reviewer for suggesting additional important references to further contextualize our work within the broader field of spintronic TRNGs and in-memory computing. Following the reviewer’s advice, we have cited the recommended seminal papers (Fukushima et al., Appl. Phys. Express 2014; Choi et al., IEDM 2014; Vodenicarevic et al., Phys. Rev. Applied 2017) in the Introduction. We explicitly clarified how our approach extends and differs from these earlier device-centric studies by highlighting our unique capability of dynamically configuring probability distributions, thus enhancing flexibility and efficiency in probabilistic algorithm applications.

Reviewer #2

Comment: The authors of “*Probabilistic Greedy Algorithm Solver Using Magnetic Tunneling Junctions for Travelling Salesman Problem*” propose a stochastic variant of the traditional greedy algorithm, utilizing the intrinsic randomness of Magnetic Tunneling Junctions (MTJs) to introduce controllable stochasticity. They employ Bayesian inference to dynamically update the Probability Density function (PDF), enabling the mapping of the binary Bernoulli TRNG outputs of MTJs to desired probability distributions. This capability is leveraged in their probabilistic greedy algorithm for decision-making during the optimization process. The authors put forth that this approach reduces complexity while improving solution quality for the NP-hard Travelling Salesman Problem (TSP). The work on using Bayesian inference for dynamic PDF generation with MTJ for TSP is particularly interesting, however, the concept itself is not novel and several aspects require clarification and further verification. Therefore, I cannot recommend it for publication in *Nature Communications*, and quite substantial improvements are necessary for further consideration.

Response: We sincerely thank Reviewer #2 for carefully reading our manuscript and providing detailed feedbacks and especially the encouraging comment “the work on using Bayesian inference for dynamic PDF generation with MTJ for TSP is particularly interesting”. While we acknowledge the concerns raised, we respectfully clarify that the presented work indeed introduces significant novelty and advancement compared to previous studies. Specifically, our paper reports a novel integration of hardware-based true randomness generated by Magnetic Tunnel Junctions (MTJs) into a probabilistic greedy algorithm framework. Contrary to the reviewer’s general assessment of

limited novelty, Reviewer #1 explicitly highlights this aspect as a “forward-looking approach” and emphasizes its “methodological novelty,” noting that our study convincingly demonstrates improved scalability and robustness compared to classical methods like simulated annealing and genetic algorithms. Reviewer #1 further commends the multi-faceted evaluation spanning device fabrication, experimental verification, and algorithm benchmarking, acknowledging the importance and distinctiveness of our contributions.

Regarding the concern about the utilization of Bayesian inference to dynamically update probability distributions, we clarify that the integration of MTJ-based hardware randomness and Bayesian updating within a probabilistic greedy solver framework has not previously been reported. Earlier studies employing Bayesian methods or MTJ-based random number generators (TRNGs) typically focused separately on either hardware-level randomness characterization or probabilistic computing at a conceptual or simulation level. Our paper, however, bridges these aspects uniquely, implementing dynamically configurable hardware-based probability distributions directly into a probabilistic greedy algorithm tailored for combinatorial optimization problems, specifically, the TSP for the first time. This experimental integration significantly advances beyond existing literatures, enabling seamless tuning of randomness and resulting in and demonstrating substantial practical performance gains — precisely the novelty and utility emphasized by Reviewer #1.

We understand the need for further clarifications and verification highlighted by Reviewer #2, and we will comprehensively address each detailed comment in the subsequent point-by-point responses. We firmly believe that once the requested clarifications are made explicit and additional supporting data or explanations are provided, the novel and substantial contributions of our work will become more transparent, alleviating the reviewer's concerns and demonstrating the manuscript's suitability for *Nature Communications*.

Question #2-1: The utilization of MTJs in solving the travelling salesman problem (TSP) is not clearly explained. I recommend that the authors provide a more detailed description. For instance, in Figure 2, the authors present random numbers with various distributions, but the connection between this aspect and the TSP is unclear. The authors should quantify and explicitly illustrate the benefits, making the advantage of this approach more apparent.

Response: We appreciate the reviewer’s insightful comment, which helped us to recognize the need to explicitly clarify the direct connection between MTJ-generated random numbers and the decision-making process of our probabilistic greedy algorithm for TSP.

In the probabilistic greedy algorithm presented in our manuscript, the key operation — selecting the next city — relies on generating random numbers from a dynamically updated probability distribution (probability density function, PDF). At each algorithmic step, the probability of selecting the next city is explicitly given by Eq. (1):

$$\begin{aligned}
 P_{i+1}(\bar{N}_i) &= (1-b_i) \exp(-d_{N\bar{N}_i}/k_B T) / Z \\
 Z &= \sum_{i=1}^8 (1-b_i) \exp(-d_{N\bar{N}_i}/k_B T)
 \end{aligned}
 \tag{Eq. (1)}$$

This clearly indicates the necessity of accurately generating random numbers according to a well-defined, step-by-step updated PDF. Here, b_i denotes the accessibility of City i for the $(i+1)^{\text{th}}$ step, $b_i = 1$ or 0 if City i has been visited or not. $d_{N\bar{N}_i}$ is the distance between City i and an unvisited one of the rest. Z is the partition function summing over all the unvisited cities and the sampling possibility $P_{i+1}(\bar{N}_i)$ of an unvisited city is dependent on the Boltzmann distribution after renormalization by Z in Eq. (1). The PDF ensures the convergence of the algorithm, the highest probability for the global optima. Actually, Eq. (1) is the broadly applied softmax function, which defines a unique PDF among unvisited cities in each step and should be configured dynamically step-by-step. This is exactly the reason why a PDF-configurable TRNG are desperately needed here.

Traditional pseudo-random number generators (PRNGs) or typical hardware random number generators lack the flexibility to dynamically and accurately alter their probability distributions in real-time, particularly across varying intermediate temperature states ($k_B T$). Our MTJ-based TRNGs uniquely overcome this limitation by enabling highly configurable random-number distributions, as already demonstrated experimentally in Figure 2. Specifically, we showed in Figure 2(b–g) that our MTJ-based system can reliably and accurately generate random numbers following different, precisely controlled distributions, including Gaussian, uniform, exponential, and user-defined arbitrary distributions.

Therefore, the capability shown in Figure 2 directly correlates with the probabilistic selection mechanism in our TSP algorithm (Figure 3). At each step, based on Eq. (1), our MTJ-based TRNG system in real time step-by-step adjusts its random-number output distribution, providing exactly the required random samples to select the next city probabilistically. This not only guarantees consistency with the theoretical PDF but also offers significant encoding efficiency, as this PDF-configurable MTJ-based approach only requires $\log_2 N$ bits of MTJs (where N is the number of cities), a significant reduction compared to classical Boltzmann-machine or Ising machine encodings that require N^2 bits of binary MTJ TRNGs. This improved encoding efficiency is also the advantage over previous ones.

To clarify this essential connection, we have explicitly modified our manuscript to include a clear statement on how the experimentally demonstrated random-number generation capabilities (Figure 2) directly feed into and enable the probabilistic greedy algorithm's city-selection process (Figure 3).

To further enhance the clarity and intuitiveness of this mechanism, we provide an additional explanatory schematic (Figure R2), which illustrates how different temperature values ($k_B T$) alter the probabilistic decision-making process in our algorithm. At low temperature, the city with the shortest distance is deterministically chosen, mimicking a classic greedy algorithm. At high temperature, cities are selected with nearly equal probability, akin to random exploration. At intermediate values, the selection probability is smoothly distributed following a Boltzmann distribution, striking a balance between exploration and exploitation.

This step-by-step probabilistic sampling is directly driven by our MTJ-based TRNGs, whose PDF can be dynamically updated to reflect the current decision state. The hardware block shown in Figure R2 (middle-bottom) highlights the compact structure needed for such TRNG-based selection: a small number of MTJs ($\log_2 N$ for N cities) suffices to implement arbitrary probability distributions at each step, thereby greatly reducing hardware complexity compared to conventional Ising-based solvers requiring $O(N^2)$ coupling nodes. This clear physical-to-algorithmic linkage — from MTJ voltage to switching probability to next-city selection — underpins the novelty and efficiency of our probabilistic greedy optimization strategy.

Figure R2. Schematic illustration of the decision-making process in the probabilistic greedy algorithm for solving TSP. Top: The influence of temperature (T) on city selection probabilities — from deterministic ($T \rightarrow 0$), to balanced probabilistic (intermediate T), to fully random ($T \rightarrow \infty$). Middle: A dynamically updated probability distribution is used to sample the next city. Bottom: This distribution is realized in hardware using MTJ-based TRNG arrays capable of generating binary outcomes according to any desired PDF, with a logic resolution of $\log_2 N$ MTJs for N cities. The overall system links real-time TRNG outputs to Boltzmann-distributed decision probabilities, efficiently guiding the optimization trajectory.

Question #2-2: The authors emphasize the use of TRNGs, but the advantage over commonly used PRNGs remains unclear. PRNGs can effectively generate distributed random numbers, and true randomness is not typically a requirement for probabilistic algorithms like this one, or even for cryptographic operations, as dedicated PRNGs are generally sufficient. The authors should justify the necessity of true randomness in this context and discuss the potential advantages of TRNGs over PRNGs in this specific application.

Response: We thank the reviewer for raising an important question regarding the advantages of true random number generators (TRNGs) over commonly used pseudo-random number generators (PRNGs) in probabilistic algorithms. Although PRNGs indeed produce sequences that approximate

randomness, their deterministic nature fundamentally limits their flexibility and real-time configurability — particularly important for algorithms relying on dynamically varying probability distributions, such as our probabilistic greedy approach.

Specifically, the key advantage of our MTJ-based TRNG over traditional PRNGs lies in its intrinsic physical randomness, enabling direct, rapid, and flexible manipulation of the generated probability distributions through external control parameters (such as voltage or pulse width). This feature is crucial for our algorithm, as the selection probability distribution for choosing the next city changes dynamically at every iteration based on updated intermediate parameters (e.g., distances and temperature $k_B T$). In contrast, traditional PRNGs cannot natively support direct, instantaneous, and precise manipulation of their output probability distributions. Achieving similar configurability with PRNGs would typically require additional computational overhead, such as mapping uniformly distributed random numbers into desired distributions using computationally intensive transformations, thus significantly reducing efficiency.

Furthermore, from a hardware perspective, MTJ-based TRNGs inherently offer parallelization and ultra-low-power operation, making them particularly suitable for energy-efficient, hardware-accelerated optimization tasks. This direct physical generation of randomness bypasses the algorithmic complexities associated with PRNG-based software transformations, thus ensuring higher performance, lower latency, and lower power consumption in practical implementations.

To clarify this point explicitly in the manuscript, we have added a concise but clear statement describing the distinct advantages of TRNGs over PRNGs within our specific application context.

We would like to highlight the difference between PRNG and PDF-configurable TRNG by using a typical figure from the perspective paper “*Probabilistic Neural Computing with Stochastic Devices*” by S. Misra et al⁵. While PRNG can be used in probabilistic computing, computing overheads in the soft level are indispensable to translate DPF from a uniform one generated in PRNG into a desired one, which is marked as Level 0 in efficiency and integrity of probability computing systems. Level 1 is more efficient than Level 0 by realizing more complex PDF in hardware than the uniform one as we have demonstrated in Figure 2 of the revised manuscript. Level 2 can achieve even higher efficiency by dynamic and parallel reconfiguration of PDF in real time according to the problem

involved, which is exactly what we have demonstrated in the probabilistic greedy algorithm solver. In this regard, our demonstration represents a remarkable advancement compared to a PRNG system.

Figure R3 of the perspective paper “Probabilistic Neural Computing with Stochastic Devices” by S. Misra et al⁵.

Question #2-3: The concept of "iteration" in the manuscript is ambiguous. How is iteration defined in this context? For example, in the case of Burma14, does the iteration count increase by one only when b1 through b14 all become 1, or does it increment with every clock cycle? Clarifying this would enhance the reader's understanding.

Response: We thank the reviewer for highlighting this ambiguity regarding the definition of "iteration" in our manuscript. We agree this terminology should be clarified explicitly to avoid confusion. In our work, the term "iteration" is defined as one complete execution of the path-

selection process, in which the algorithm sequentially selects each city exactly once until all cities have been visited, forming a complete route (for example, selecting from city 1 to city 14 exactly once in the Burma14 problem constitutes a single iteration). Thus, each "iteration" refers to the generation and evaluation of one full TSP solution (route), rather than a single incremental selection step or a clock cycle.

To clearly communicate this definition to readers, we have explicitly clarified this terminology at the first occurrence of the term "iteration" in the manuscript. We would like to thank the referee for pointing out this critical issue.

Question #2-4: The probabilistic greedy algorithm proposed by the authors closely resembles the epsilon-greedy search widely used in reinforcement learning and machine learning. Epsilon-greedy methods strike a balance between exploration (randomness) and exploitation (greedy optimization). There is no acknowledgement of these existing methods, as epsilon greedy also combines randomness into the greedy algorithm with a Q learning parameter. There exists literature on these techniques and an instance of TSP implementation as well in "doi.org/10.1049/tje2.12303", the details of which are missing. The authors do not discuss how their approach is unique or the impact of dynamic PDF generation with MTJs.

Response: We sincerely appreciate the reviewer's critical observation regarding the relationship between our proposed probabilistic greedy algorithm and the widely known epsilon-greedy methods employed in reinforcement learning and machine learning. We acknowledge that both epsilon-greedy approaches and our probabilistic greedy algorithm share the core principle of balancing exploration (randomness) and exploitation (greedy optimization). Nevertheless, we would like to emphasize the distinct differences between them and unique contributions of our work clearly:

- 1) The traditional epsilon-greedy method typically introduces randomness through a fixed parameter (ϵ), determining the probability of selecting a random choice rather than the optimal (greedy) choice at any step. In contrast, our approach dynamically adjusts the probability distribution for selecting the next city at each step. Specifically, the probability of selecting each unvisited city is continuously updated according to a precise and strict mathematical form (exponential dependence on city distance and temperature parameter

$k_B T$, Eq. (1)). Thus, our algorithm inherently provides a more sophisticated, fine-grained, and adaptive form of exploration, compared to the static binary "greedy vs. random" decision-making characteristic of epsilon-greedy methods. Worth noting, the dynamic PDF defined in Eq. (1) is delicately designed to force trial solutions to obey the Boltzmann distribution – the lowest route having the highest probability especially at lower temperatures, which ensures the effectiveness and convergence of the algorithm and usefulness of the simulated annealing strategy. This is the reason why the PDF at each step should be dynamically modified and also the reason why a PDF-configurable TRNG matches the need of this probabilistic greedy algorithm.

- 2) Another major distinction is our direct integration of hardware-level randomness provided by MTJ-based TRNGs, enabling real-time, precise, and computationally efficient generation of dynamically configured probability distributions. While traditional epsilon-greedy methods rely solely on software-generated pseudo-randomness (uniform distribution), our MTJ-based TRNG system enables direct sampling from tailored, arbitrary probability distributions. This intrinsic flexibility allows our algorithm to immediately and precisely adapt its exploration-exploitation strategy in response to changing algorithmic conditions, significantly enhancing its performance and scalability, as also indicated by above Figure R2.
- 3) From an implementation perspective, our MTJ-based approach also drastically reduces hardware complexity, requiring only $\log_2 N$ probabilistic bits to represent and select among N possible cities. Traditional epsilon-greedy or related methods, typically relying on uniform PRNG-based sampling, would require additional computational transformations or overhead to achieve similar distributional flexibility, thus potentially limiting efficiency and scalability.

We agree with the reviewer's suggestion of explicitly discussing these distinctions, clearly acknowledging existing epsilon-greedy methods and highlighting the unique advantages of our dynamic PDF generation strategy via MTJ-based hardware randomness. We have therefore explicitly included references to existing epsilon-greedy methodologies (including the reviewer's suggested reference: doi.org/10.1049/tje2.12303), clarified the conceptual differences, and

explicitly described how our MTJ-based approach uniquely enhances the exploration-exploitation trade-off within probabilistic greedy algorithms.

Question #2-5: There are specific details on their measurement setup but not on the MTJ devices they are using. Like variation in the patterned device dimension, the switching curve deviations of MTJs, cycling degradation and the impact of temperature changes on device switching characteristics. Also, what methods are employed to mitigate if such issues are present? The sigmoid switching probability curve should be discussed for all the devices being used to solve the TSP problem and mention the number of devices used to solve the problem.

Response: We thank the reviewer for highlighting the need to explicitly discuss the potential device-level variations and operational challenges of the MTJs utilized in our experiments. We recognize that understanding device-level variation — such as patterning dimension variability, switching curve deviations, cycling-induced degradation, and sensitivity to temperature—is critical to ensuring stable and reliable random-number generation performance, especially in practical scenarios involving multiple MTJs.

In response to this concern, we emphasize that our research group recently conducted an in-depth experimental and numerical study specifically addressing the robustness and stability of MTJ-based true random number generators under realistic operational conditions, including temperature-induced drift, device aging, and environmental instability⁴. In this recent work, we proposed and experimentally validated a hybrid control strategy combining self-stabilizing feedback loops and pulse-width modulation schemes. Notably, we developed a "downcalibration-2" approach, which efficiently updates control parameters at two-step intervals using simple integer-resolution timing logic. This method ensures excellent statistical stability of the generated random numbers without external calibration, bit discarding, or extensive pre-characterization processes.

The MTJ devices utilized in the current study have a similar structural and fabrication setup as reported in the previous work. Therefore, they naturally inherit the demonstrated robustness against common device variations and operational drift issues. Specifically:

- 1) Our self-stabilizing feedback loop dynamically corrects switching probability deviations caused by minor variations in patterned device dimensions or switching curves. Such

device-to-device variability is automatically compensated through the hybrid control strategy, ensuring consistent output characteristics across multiple MTJ units.

- 2) The pulse-width modulation component of our hybrid scheme efficiently handles potential cycling-induced degradation, continuously adapting pulse timing parameters to maintain stable switching probabilities even after extended device operation.
- 3) Extensive experimental validation under dynamic temperature conditions confirms that the employed hybrid stabilization strategy robustly mitigates thermal fluctuation impacts, maintaining stable MTJ performance over a wide operational temperature range (as detailed explicitly in Ref.⁴).

Regarding device numbers and characterization details used in the current TSP experiments, we clarify explicitly in the revised manuscript: A small number of MTJs (typically $\log_2 N$, e.g., four MTJs for the Burma14 problem) were utilized, with the switching probabilities of all devices characterized and confirmed to follow closely aligned sigmoid probability curves as shown below in Figure R4. The devices' statistical switching properties and consistency were validated prior to use in algorithmic experiments.

Figure R4. The P - V characteristics of the used 4 MTJs in the probabilistic greedy algorithm. As mentioned above, these device-to-device variation and long-term drift of the performance of a device can be compensated by our recent developed methodology as uncovered in Ref.⁴ explicitly.

We have now explicitly referenced this prior comprehensive work in our manuscript, clearly stating that device-level variations and operational instabilities are effectively mitigated by the previously validated hybrid control methods, thus substantiating the reliability and consistency of the MTJs employed here.

Question #2-6: The voltage control as discussed by the authors is highly precise and stabilized with feedback as mentioned in the supplementary. However, there are concerns about how such precision will be maintained when scaling the system to address larger problems. When transitioning to on-chip implementations, maintaining precision will require the integration of high-resolution ADCs and DACs impacting overall system performance.

Response: We appreciate the reviewer's thoughtful concern regarding the scalability of our system and the challenge of maintaining high-precision voltage control when transitioning from laboratory setups to on-chip implementations. This is indeed a critical consideration for practical deployment of MTJ-based TRNG systems in large-scale hardware-accelerated computing.

To address this challenge, our recent work⁴ directly tackles the issue of voltage precision and stability in a digital system. In that study, we proposed and experimentally validated a hybrid control strategy that replaces the need for ultra-precise analog voltage sources or high-resolution DACs with digital pulse-width modulation (PWM). Rather than finely adjusting voltage amplitude, which often demands costly and power-hungry high-resolution DACs, our approach modulates the duration of fixed-amplitude pulses using simple integer-timed digital logic. This significantly simplifies the hardware DAC and ADC requirements for large-scale implementations and ensures that precise control of switching probability can be achieved even on resource-constrained platforms.

This PWM-based approach was shown to be highly effective in maintaining target probability distributions under both device variations and environmental fluctuations, including temperature drift, while being inherently more scalable and integrable into CMOS-compatible architectures. As

a result, our probabilistic greedy algorithm and TRNG architecture can be feasibly extended to large-scale and on-chip systems without relying on high-resolution analog components.

Besides, the developed probabilistic greedy algorithm has a high encoding efficiency, only $\log_2 N$ bits of MTJs being used to encode a N -city TSP, which also mitigates hardware complexity when scaling to larger N -city TSP.

We have added a summary of this result to the manuscript and clarified that our approach is intentionally designed with hardware scalability in mind, leveraging digital timing rather than analog precision to maintain performance.

Question #2-7: The manuscript lacks a detailed evaluation of the randomness and entropy tests for the TRNG-generated random bitstream data. The authors just briefly touch upon the neighboring correlations of the data generated out of the Bayesian inference. It is also unclear whether the MTJs employed in their setup require postprocessing of the random bitstreams due to device variations, control and environment impacts that alter the device switching characteristics. The speed of random bitstream generation has not been explicitly addressed. Although they mention in their previous work on SOT-MTJ [32] that they pass the randomness tests with post-processing, there is no such discussion in this work.

Response: We thank the reviewer for highlighting the need to elaborate on the statistical evaluation of the TRNG-generated bitstreams, including randomness quality, entropy, post-processing, and speed. These are crucial aspects for any TRNG-based system, and we appreciate the opportunity to clarify.

In this study, the MTJs used for TRNG were fabricated and tested using the same experimental protocols as those detailed in our recent publication⁴. In that work, we performed an extensive evaluation of the random bitstreams using NIST SP800-22 statistical test suite. The MTJ-based TRNG passed all tests with high confidence without requiring post-processing, thanks to the implementation of a hybrid control strategy that actively stabilizes switching probabilities across environmental and temporal variations. This demonstrates that the raw bitstreams generated by our MTJs already exhibit high-quality randomness and entropy, eliminating the need for whitening or post-processing in most use cases.

Figure R5. The MTJ-based TRNG passed all NIST tests without requiring post-processing, under Downcalibration-2 strategies.

In the current manuscript, while we have primarily focused on demonstrating the tunable probabilistic distribution for algorithmic integration, we have now added a brief summary of the statistical randomness validation from our previous work and clarified that the same devices and testing protocols were employed.

Regarding speed, the current bit generation rate is limited primarily by the bandwidth of the data acquisition system (currently approximately 500 kHz per MTJ using our NI PXIe setup). However, the intrinsic switching dynamics of STT-MTJs permit much faster operation. With appropriate peripheral circuitry and high-speed readout electronics, generation speeds approaching GHz rates have been demonstrated, as reported in recent works such as arXiv:2501.06318⁶. We have now included this discussion in the manuscript to clarify both the current setup and future performance potential.

Question #2-8: The conversion of binary Bernoulli TRNG to output based on any PDF, especially Gaussian distribution using the Bayesian inference is a versatile approach. The distribution is used as the prior and based on conditional probability the voltages to the MTJ are controlled, generating random numbers in the required probability distribution. The authors outline this already in their

previous work in reference [32]. While dynamically computing the PDF for TSP is a good strategy, the conditional probability limits the speed and control of the MTJs as they depend on the outcome of the previous node. The authors need to address and clarify this issue.

Response: We thank the reviewer for this insightful observation regarding the potential impact of conditional probability structures on speed and control in the PDF conversion process. We also appreciate the recognition that dynamically computing the PDF for TSP is a valuable strategy.

It is true that conditional probability modeling — particularly in Bayesian inference — can introduce data dependencies that may impact speed, especially when later stages depend on the outcomes of earlier nodes. In our framework, such conditionality is indeed present when generating non-uniform distributions, as each city-selection step depends on the outcome of previous steps. However, this dependency is relatively shallow (i.e., step-by-step rather than deeply nested), and is handled using synchronous clocked updates.

Specifically, the MTJ control voltages or pulse widths are configured at each step based on the current problem state, which includes the previously visited cities. While this introduces some degree of sequential operation, the update process is lightweight and does not involve complex inference trees or deep conditional chains. Moreover, the update and sampling process are implemented as synchronized pipelined operations, meaning that the latency is limited to a fixed number of clock cycles per decision step.

As demonstrated in our recent work [Zhang et al., PR Applied 23, 054073 (2025)]⁴, the control logic required to generate these distributions can be efficiently implemented using digital circuitry and pulse-width modulation (PWM), enabling fast and accurate mapping from Bernoulli bits to arbitrary PDFs with minimal resource overhead.

Therefore, while conditionality is inherent to the stepwise decision-making process in TSP, it does not significantly restrict speed in our implementation, especially considering that this conditionality is further mitigated by the $\log_2 N$ encoding complexity. This is because the sampling does not rely on recursive inference or variable-depth logic, but instead on fixed-control updates derived from simple heuristics. We have clarified this aspect in the revised manuscript to better convey how our implementation balances conditional logic and runtime efficiency.

Question #2-9: The voltage precision plays a critical role in achieving the desired probability distributions, and as the problem scales the requirement becomes more stringent. Conditional probabilities that are close in value demand highly precise and stable voltage adjustments, along with a repeatable sigmoidal switching trend. The shape and consistency of the sigmoid curve are crucial for accurately determining the resulting PDF, but variations in device characteristics can significantly impact this curve. There is no discussion on how the authors plan to address such issues when scaling, as they currently use precision instruments to apply the voltage.

Response: We appreciate the reviewer's concern regarding the precision and stability of voltage control, which is indeed critical for achieving accurate and repeatable probability distributions, especially when conditional probabilities are closely spaced. Additionally, we fully agree that the shape and reproducibility of the sigmoid switching probability curve play a central role in enabling high-fidelity probability distribution generation using MTJs.

To address these issues, we refer to our recent work⁴, in which we investigated and resolved this challenge through a hybrid control strategy. Instead of relying on high-resolution voltage amplitude tuning — which, as the reviewer correctly notes, becomes increasingly impractical with scale — we implemented pulse-width modulation (PWM) to control the switching probability. In this approach, the amplitude of the applied voltage remains fixed, and only the duration of the pulse is adjusted using simple digital logic. This method allows for fine-grained control over the effective switching probability while maintaining the sigmoidal characteristic of the MTJ response as experimentally shown in Figure R4.

Furthermore, our work demonstrated that this PWM-based control, in combination with a self-stabilizing feedback loop, results in consistent sigmoid behavior across multiple devices, despite inevitable variations in fabrication or temperature. The feedback mechanism periodically monitors and corrects the effective switching behavior using only integer-resolution timing updates, removing the need for external high-precision instrumentation or costly pre-characterization.

This architecture was shown to maintain stable and accurate probability distributions even under thermal drift and aging, making it robust and scalable for hardware-accelerated probabilistic computing. In the current work, we employ the same MTJ structure and control scheme, and thus

inherit the same robustness and precision benefits. We have clarified this point in the revised manuscript to emphasize that our method remains reliable and scalable even in the presence of close-valued conditional probabilities.

Question #2-10: The authors describe that the $k_B T$ value is empirically found for optimal balance between exploration and exploitation. However, this implies that $k_B T$ would need to be tuned for every specific problem instance or whenever other hyperparameters of the algorithm are modified. The authors should provide more discussion on how this parameter is affected. One can see that $k_B T$ is 60 in the burma14 a 14 city TSP and 1 in the st70 a 70 city TSP suggesting significant variability in its value. Then this search would be similar to the Boltzmann machine-based SA, where each temperature step would govern a local search. Instead of the algorithm finding one on its own, here the optimal temperature is predetermined empirically. Finding this optimal temperature would then become a problem on its own. I suggest that the authors provide a theoretical framework or methodology to determine the optimized $k_B T$ for different scenarios.

Response: We thank the reviewer for the astute observation regarding the role of the $k_B T$ parameter in governing the balance between exploration and exploitation within our probabilistic greedy algorithm. Indeed, this temperature-like parameter plays a critical role in shaping the dynamic probability distribution used for city selection and can strongly influence the quality of the resulting solution.

We fully agree that the current implementation selects $k_B T$ empirically for each problem instance, which introduces a hyperparameter tuning overhead and may influence the generality of the method. This behavior does bear similarity to the temperature schedule in Boltzmann machines and simulated annealing frameworks, where the temperature controls local search characteristics. The parallel is well-taken.

At present, we lack a fully developed theoretical framework for analytically determining the optimal $k_B T$ for arbitrary problem instances, which is generally regarded as a tough question for any probabilistic computing algorithms. However, in practice, we have observed a consistent heuristic: the optimal $k_B T$ tends to correlate with the scale of the distance values in the TSP problem. In our workflow, we often begin with a rough estimate based on the average or one-tenth of the maximum

pairwise distance in the TSP cost matrix, and perform a few fast exploratory runs to converge on an effective value. While we acknowledge that it does not constitute a principled solution, this empirical approach has proven effective across a range of tested problem sizes.

We appreciate the reviewer's suggestion and agree that establishing a theoretically grounded or adaptive strategy for selecting $k_B T$ remains an important direction for future work. One possible route is to frame the tuning of $k_B T$ as an outer-loop optimization problem or to integrate temperature adaptation heuristics drawn from thermodynamic-inspired algorithms. This remains an open and valuable challenge in developing fully autonomous probabilistic solvers.

We have added a corresponding note in the discussion section of the revised manuscript to acknowledge this limitation and outline our future efforts to address it.

Question #2-11: Figure 4(c) and (e) represent the density of solutions, and it is unclear how many solutions form the basis of these density metrics. The metrics can be discussed in terms of the probability of a solution for better clarity. There is no mention of the time taken to solve these TSP instances or any insight into the trend of time to solution as the problem size increases. Only algorithm space and time complexities are provided. The actual time required by their hardware to solve the problem is not discussed. The analysis of the energy and area occupied by their system is also vital, which is absent.

Response: We thank the reviewer for pointing out the lack of detail regarding solution density quantification, runtime evaluation, and system-level metrics such as energy and area.

First, we clarify that the density plots in Figure 4(c) and 4(e) are based on 100 independent runs of the algorithm for each problem instance. These were used to generate a histogram of the frequency of solution energies, giving a probabilistic view of the algorithm's convergence behavior. We have added this detail to the figure caption for clarity.

Regarding the runtime and scalability, we agree that providing absolute runtime is valuable. However, our current setup is an experimental hardware-in-the-loop prototype designed for proof-of-concept, using a NI PXIe system for MTJ voltage control and data acquisition. The time to generate a solution in this setting includes latency due to instrument communication, software overhead, and sequential device control — not representative of optimized hardware

implementations. Instead, we would like to use iteration numbers of samplings, a hardware-weakly dependent parameter, as the time-marker to show the convergence speed.

Here, to give an intuitive estimation, we can achieve an iteration of samplings for the Burma 14 TSP within 0.2 s based on our recent measurement system. Stressing again, a fully hardware solver should achieve much higher speed.

That said, the MTJ switching time itself is in the order of nanoseconds, and our recent work⁴ demonstrated reliable switching at rates consistent with several hundred megahertz. Further, recent MTJ-based TRNG systems in literature⁶ demonstrate generation rates approaching the gigahertz level. This suggests that a fully integrated system could yield solutions orders of magnitude faster than our current prototype.

Concerning energy and area, we fully acknowledge that these are important metrics for evaluating hardware feasibility. However, this work primarily focuses on demonstrating the algorithmic potential of a probabilistic greedy solver using spintronic randomness. A rigorous energy and area evaluation would require a fully integrated ASIC or FPGA implementation, which lies outside the scope of this proof-of-concept demonstration. Nevertheless, we expect substantial benefits in scalability and energy efficiency due to the inherently low power consumption and non-volatility of MTJs, as supported by prior studies.

We have added clarification on the number of trials, explained the nature of the prototype system, and discussed the need for future integrated design exploration to address energy and area quantitatively.

Question #2-12: The authors have solved st70 TSP by simulating the probabilistic greedy search algorithm. The authors of reference [36] have implemented a similar st70 problem on hardware. With simulation data provided by the authors, a proper comparison cannot be drawn on the performance of the proposed approach. Factors of voltage precision levels or the number of available hardware MTJ devices could restrict larger-scale implementation like the 70-city problem in the probabilistic greedy methods with MTJs. Benchmarking different methods in terms of hardware performance is essential to substantiate the main claim. A scalability analysis should be

included, showing the computational costs and performance metrics as the number of cities increases.

Response: We appreciate the reviewer’s insightful comment regarding the limitations of simulation-based evaluation for large-scale problems such as ST70, and the importance of benchmarking hardware performance to support claims of scalability.

Indeed, our current implementation of the ST70 instance uses simulation to evaluate the algorithmic behavior and validate that the probabilistic greedy strategy maintains high solution quality and robustness as the problem size increases. The hardware demonstrations in this work — including the Burma14 example — serve as proof-of-concept validations of the core MTJ-based randomness engine and its integration into probabilistic decision-making processes.

We fully agree that demonstrating large-scale deployment using actual MTJ arrays is a critical next step. Worth mentioning, our method is inherently more scalable than previous Boltzmann or Ising machine architectures because it only requires $\log_2 N$ MTJs to encode a probability distribution over N choices, as opposed to the latter architectures that require N^2 MTJs to encode an N -city path. This logarithmic scaling allows for significantly more compact and energy-efficient implementations in future chip-level deployments.

As noted in Ref.⁴, our pulse-width modulated control scheme further supports this scalability by eliminating the need for high-resolution DACs. Therefore, we believe our approach is well-suited for large-scale hardware realization, but acknowledge that a full chip-level implementation and layout-aware parallelization strategy will be our future direction to rigorously validate scalability in terms of area, energy, and performance.

Benchmarking across different hardware platforms remains challenging due to differences in architectural assumptions and design goals (e.g., parallel annealing-based arrays vs. sequential probabilistic decision chains). Nevertheless, we have added a discussion in the revised manuscript outlining the future roadmap for scalability validation and comparative benchmarking under unified performance metrics.

Question #2-13: The information presented in Figure 5 is insufficiently detailed. The subfigures of Fig. 5 should also be cited properly in the main text. In Figure 5b, the axes lack numerical values,

and concepts such as time and space complexity are not adequately defined. In Figure 5d, the comparison details are insufficient — are optimal parameters used for the alternative methods? There is only block diagram in Fig. 5e. What will the system performance be expected in Fig. 5e? Some quantitative analysis would be required. While the authors claim superior performance in solution quality and convergence speed compared to classical methods, this is not convincingly demonstrated. I further recommend that the authors scale up the number of cities incrementally using different methods and present the results to better illustrate the performance of their proposed method, and provide a quantitative comparison of their algorithm performance with other state-of-the-art approaches, particularly in terms of solution quality and scalability. Benchmarking against state-of-the-art combinatorial optimization problem (COP) solvers would also help establish the novelty and effectiveness of the proposed algorithm.

Response: We thank the reviewer for their detailed and constructive feedback on Figure 5 and the associated analysis.

Regarding Figure 5b, we clarify that this is a theoretical schematic used to qualitatively illustrate the relative scaling behavior of algorithmic complexity and randomness control. As such, precise numerical values are not included on the axes, but the figure is drawn proportionally to reflect known theoretical trends (e.g., logarithmic vs. linear vs. exponential scaling). We have added a clarification in the figure caption and main text to avoid confusion.

For Figure 5d, we confirm that the comparison to simulated annealing (SA) and genetic algorithm (GA) was performed using standard parameter settings commonly used in the combinatorial optimization literature. In particular, for SA we used an exponentially decaying temperature schedule with an initial temperature proportional to the average edge weight; for GA, we used a population size of 100 and a mutation rate of 0.1. All methods were evaluated using 100 independent runs to ensure statistical reliability. These details have now been included in both the figure caption and the main text.

Figure 5e serves as a conceptual architectural diagram of a future MTJ-based probabilistic computing system. While performance data for this architecture are not available at this stage, we now provide estimated throughput projections in the text, based on our experimental switching rate

(~500 kHz) and existing literature showing MTJ-based TRNG operation exceeding 1 GHz. These estimates suggest that high-speed performance is attainable with modest parallelization in future integrated platforms.

We agree with the reviewer that more systematic benchmarking and scalability evaluation will be critical to validate the full potential of our approach. To this end, we have included a discussion outlining our roadmap for future work involving cross-platform comparisons with established solvers such as Concorde, LKH, and neural combinatorial optimization frameworks.

Lastly, all subfigures in Figure 5 are now explicitly cited and discussed in the revised main text.

Minor comment #2-1: The paper mentions " n " in the context of scalability but does not define it clearly upon first usage.

Response: We thank the reviewer for pointing this out. In the revised manuscript, we have explicitly defined n as the number of cities (nodes) upon its first mention in the context of scalability and algorithmic complexity.

Minor comment #2-2: The statement that "multiple MTJs are connected" lacks specificity regarding the exact number of MTJs.

Response: Thank you for pointing out the lack of specificity. We have clarified the number of MTJs used in our implementation: for a problem with N cities, we use approximately $\log_2 N$ MTJs to represent the required probability distribution. In practice, a small number of MTJs (e.g., 4 MTJs for 14 cities) are reused through time-multiplexing to reduce hardware overhead. This information has been added to the revised manuscript where the MTJ connection is first discussed.

Minor comment #2-3: Fig. 1e shows a switching characteristic with respect to time similar to superparamagnetic MTJ, which is against a stable STT-MTJ behavior. This confusion should be addressed.

Response: We thank the reviewer for pointing out this potential confusion. We confirm that our experiments use thermally stable STT-MTJs with perpendicular magnetic anisotropy (PMA), operating near their critical switching thresholds. The stochastic switching behavior shown in Fig. 1e corresponds to this sub-threshold regime, where probabilistic switching is induced by thermal

activation under finite voltage pulses — not similar to the behavior of superparamagnetic MTJs. No changes were made to the manuscript, as the distinction is already implied by our discussion of STT switching thresholds and device parameters, but we appreciate the opportunity to clarify this point.

Minor comment #2-4: Measuring only the neighboring correlation of generated random numbers is insufficient to demonstrate statistical independence or high-quality randomness. The authors should validate the quality of their random numbers by testing the entire sequence using standard statistical tests.

Response: We appreciate the reviewer's suggestion. We agree that evaluating statistical independence and entropy over the full bitstream is critical for assessing the quality of TRNGs. In this work, we reused the same STT-MTJ devices and readout setup as in our recent publication⁴, where comprehensive randomness testing — including full-sequence NIST SP800-22 and autocorrelation analysis — was conducted and passed without requiring post-processing as also shown in Figure R5. To avoid redundancy, we did not repeat the full statistical tests here, but have now clarified this point in the manuscript.

Minor comment #2-5: Figure 4(b) explanation on Page 8 of the main text has an error in identifying the curves.

Response: We thank the reviewer for catching this oversight. We have carefully re-examined Figure 4(b) and the associated explanation in the main text. The curve identification in the original version was indeed incorrect, and we have now corrected the description to accurately match the legend and data shown in the figure. The revised manuscript reflects this correction.

Minor comment #2-6: In Figure 4(e) what is Density (best*50), is it a distribution plot of solutions with 0 kilometers of the known optimal as mentioned on Page 8 of the main text? And the numbers on the peak are the maximum number of solutions giving exact distances? More clarity is required.

Response: We appreciate the reviewer's request for clarification regarding Figure 4(e). The term "Density (best \times 50)" refers to the relative occurrence frequency of optimal solutions (i.e., solutions with the same length as the known optimum). Since the absolute probability of obtaining the optimal solution is relatively low, this value was scaled by a factor of 50 for better visibility. The numbers

shown at the peaks indicate the corresponding temperature values ($k_B T$) that yield the highest density of optimal solutions. We have clarified this in both the figure caption and the main text.

Minor comment #2-7: Figure 5(d) shows the results of different heuristic algorithms with the error bars, there is no mention of how many samples is the error bar drawn. Prob-Greedy data is not present for 10^5 iterations in solving st70 on simulation?

Response: We thank the reviewer for this observation. The error bars in Figure 5(d) reflect the standard deviation 10 independent runs for each algorithm and each iteration setting.

As for the absence of the Prob-Greedy data point at 10^5 iterations for the st70 instance, this was due to simulation time constraints (10 hours currently for 10^5 iterations), since each run at such a large scale becomes computationally intensive. However, the trend shown up to 10^4 iterations already demonstrates the superior performance and early convergence behavior of the Prob-Greedy approach compared to the other heuristics. We believe the existing results can sufficiently illustrate the effectiveness of our method, and we have clarified this point in the revised manuscript and the figure caption.

Minor comment #2-8: In the description and main text explanation of Figure 4(d) optimal solution with 1000 repetitions is used, but in the graph the axis says iterations. Are repetitions and iterations used interchangeably? Or are the authors performing 1000 repetitions of the iterations?

Response: We thank the reviewer for highlighting this terminology inconsistency. In the context of Figure 4(d), the terms repetitions and iterations are used interchangeably to refer to one full execution of the algorithm that generates a complete TSP path solution. To avoid confusion, we have revised the text and figure caption to use a consistent term — “iterations” — and clarified that in this figure, it denotes the number of repeated runs of obtaining an individual complete TSP route.

Reviewer #3

Comment: The manuscript introduces a probabilistic greedy algorithm based on stochastic Magnetic Tunnel Junction (MTJ) in the framework of probabilistic computing. Through dynamically modulating the degree of randomness, the algorithm herein has manifested the

performance in combinatorial optimization problems. The authors claim that, when applied to the Traveling Salesman Problem, this algorithm surpasses other classical approaches, characterized by the solution quality and more rapid convergence speed.

Response: We thank the reviewer for their concise summary and for recognizing the coherence of our work. We would like to clarify that while probabilistic optimization is a known concept, our study presents a distinctive contribution by implementing it directly through the intrinsic stochasticity of magnetic tunnel junctions (MTJs). Unlike conventional algorithmic randomness, the degree of stochasticity in our approach is continuously tunable via direct voltage control at the hardware level. This enables real-time adjustment of exploration and exploitation behavior without relying on pseudo-random sequences or external calibration. Furthermore, the integration of Bayesian inference network to map binary MTJ outputs to dynamically reconfigurable probability density functions (PDF) establishes a novel and efficient mechanism to guide the optimization process in a fully hardware-compatible manner. We believe these combined features, validated both experimentally and through large-scale simulations, distinguish our work from existing probabilistic solvers.

Comment: Overall Evaluation: This work is overall coherent. However, it does not meet the high standard of *Nature Communications* due to the lack of both novelty and enough new results:

Response: We respectfully appreciate the reviewer's "overall coherent" assessment, but we would like to emphasize that the presented work introduces a new combination of hardware-level true randomness and adaptive probabilistic control for solving combinatorial optimization problems. While MTJ-based TRNGs and greedy algorithms have been studied separately, our work is the first to link them in a scalable and tunable optimization framework that is experimentally validated on a physical MTJ platform. The ability to dynamically adjust the probability distribution based on system state through hardware-controllable parameters offers a flexible and low-overhead solution that has not been demonstrated in this context. We have revised the manuscript to better articulate this novelty and to clarify the broader impact of our approach for energy-efficient, hardware-embedded computing.

Question #3-1: The key experimental finding in this work is that four asynchronous MTJs are utilized to generate the desired probability distribution function (Figure 2). These results have already been reported and well-studied in a previous paper by the same group [Advanced Science, 11(23) 2402182 (2024)]. The probability distributions are the same, just with a different set of measurement data.

Response: We thank the reviewer for pointing this out. While it is true that the general principle of mapping MTJ-based binary outputs to arbitrary probability distributions was introduced in our earlier work [Adv. Sci. 11(23), 2402182 (2024)], the present study involves a significantly improved experimental setup with enhanced device stability, signal-to-noise ratio, and control resolution. As a result, the newly acquired data used in this work exhibit substantially higher statistical quality, which is critically indispensable for validating the performance of the probabilistic greedy algorithm introduced here. In particular, the ability to precisely tune and reproduce the desired probability distribution is directly linked to the algorithm's effectiveness in solving the TSP. Therefore, although the underlying principle of the PDF-configurable TRNGs remains consistent with our prior publication, it is indispensable to directly demonstrate in experiment the effectiveness of more accurate PDF distributions from upgraded hardware within this newly-developed algorithmic context – the probabilistic Greedy algorithm context with the Boltzmann distribution – which just needs the PDF-configurable TRNG.

The impact of direct demonstration in matchingness between algorithm and hardware can be also indicated by the perspective review by Misra⁵ et al. In this article, according to the degree of integration between software and hardware and as-caused computing efficiency, probabilistic computing systems are classified into 4 levels.

- In Level 0, TRNG hardware only generates random numbers according to the uniform distribution; the desired PDF by algorithm can only be further transformed from the uniform distribution with extra computing overheads and less sampling efficiency, for example, via the Acceptance-Rejection Sampling Method.
- In Level 1, the desired or predicted PDF by an algorithm can be directly sampled from TRNG hardware. The sampled random numbers are fed back further into the algorithm. This

application scenario is broadly encountered, for example, in many Digital Twin Systems where such typical PDFs as the Gaussian or exponentials or others are widely adopted. This kind of TRNG hardware, especially with PDF-configurability, is reported by our Adv. Sci. paper.

- In Level 2, an algorithm in a solver, for example, of solving COPs, predicts problem-specific or even state-specific PDFs to improve correctness and efficiency. In this case, PDFs can be in any arbitrary forms and should be updated in real time; thus PDF-configurable TRNGs are indispensable, among which the Boltzmann or Ising machines are typical. Via the local Gibbs sampling protocol, the tunable PDF is constrained into binary or Bernoulli one, which can be directly sampled from MTJ hardware. However, the cost of this binarization or algorithm is low encoding and sampling efficiency (N^2 for N -city TSP). In order to substantially improve efficiency, more efficient algorithms are highly desirable, which accordingly call for a more general PDF-configurable TRNG hardware beyond the binary one.
- In Level 3, a full hardware solver is anticipated with all the other components besides of stochastic samplings also implemented in hardware. In our case, the other calculation overheads include addition, multiplications and exponentials; they can be realized in MCU or via look-up-tables, which should not impose a fundamental obstacle by an incremental improvement based on Level 2.

The novelty and significance of our work naturally emerges from this architecture: Our work, proposing the efficient probabilistic greedy algorithm for TSP ($\log_2 N$) and experimentally demonstrating the matchingness between the algorithm and PDF-configurable MTJ-based TRNG hardware, just hits the goal of Level 2.

Figure R3 of the perspective paper “*Probabilistic Neural Computing with Stochastic Devices*” by S. Misra et al⁵.

Question #3-2: (2) The probabilistic greedy algorithm highlighted in this work is also not new. The concepts and algorithms have been previously reported in several papers many years ago [Computers & Operations Research, 37: 432 (2010); IEEE Transactions on Communications, 581: 3286 (2010); Proceedings of the 42nd IEEE International Symposium on Multiple-Valued Logic (ISMVL), Victoria, CANADA, F 2012 May 14-16, (2012)].

Response: We thank the reviewer for pointing out the earlier works on probabilistic variants of greedy algorithms. Indeed, the idea of introducing randomness into greedy decision-making has a rich history. However, our approach substantially differs both in its theoretical foundation and its physical realization.

Table R1. Comparison of our probabilistic greedy algorithm with typical pioneering others. Essentially, they differ in the final distribution of trial solutions, the Boltzmann style or not.

Feature	Our Work	C&OR, 2010	IEEE TCOM, 2010	ISMVL 2012
Target Problem	TSP	TSP	Channel Assignment in Cellular Networks	Circuit Mapping & Logic Assignment
Greedy Structure	City-by-city selection with Boltzmann-shaped probability	Deterministic greedy with randomized restart	Greedy with probabilistic selection using ranking weights	Greedy logic selection perturbed with random choices
Probability Foundation	Explicit Boltzmann distribution over distance cost , dynamically updated	Simple uniform/random choices or restarts	Weighted random choice based on scheduling heuristics (π, θ values)	Stochastic gate mapping heuristics
Temperature Parameter ($k_B T$)	Core parameter balancing exploration vs. exploitation, tunable	Not used	Not used, implicit preference via h-best ranking	No physical temperature analog
Path Probability	Global solution probability follows full Boltzmann distribution	No formal probabilistic model over entire path	Local decision probability based on heuristic scores	Localized randomized perturbation, no full PDF definition
Final-Step Correction	Yes: penultimate city selection includes round-trip return	No	Not applicable	Not applicable
Hardware Integration	Directly sampled from MTJ-based TRNGs with voltage/PWM control	Software simulation only	Software simulation only	Hardware-suggestive only, not implemented
Random Number Type	True Random Numbers (TRNG) from MTJs, with configurable PDFs	Implicit PRNG	Weighted pseudo-random choices	PRNG-based heuristics
PDF Configurability	Fully arbitrary distribution generation verified	Not considered	Heuristically defined probabilities	Not considered
Encoding Efficiency	Requires only $\log_2 N$ MTJs for N cities	Not addressed	Not applicable	Not addressed
Novelty Claim	First to combine Boltzmann-based probabilistic greedy with hardware-based PDF-matching TRNG	Algorithmic variant only	Algorithmic variant with diversification	Logic-oriented variation

Experimental Realization	Yes — MTJ-based setup, tested in 14-city demo	No hardware implementation	No hardware implementation	Not hardware-tested
---	----------------------------	----------------------------	---------------------

Specifically, the probabilistic greedy algorithm in this work is grounded upon the Boltzmann distribution, where the selection probability for each candidate node is determined by an energy function proportional to the total path distance as shown in Eq. (1). We also introduce a correction at the penultimate node to ensure proper termination, which forces the entire trial solutions to obey to the Boltzmann distribution over global distances. To the best of our knowledge, such a formulation — where the global path cost distribution matches the Boltzmann form and is meanwhile experimentally sampled through MTJ-based hardware — has not been previously reported. Though the Boltzmann or Ising machines also realize the Boltzmann distribution, their encoding efficiency N^2 is far different from the $\log_2 N$ efficiency of this probabilistic greedy algorithm. All these effectiveness and efficiency of the algorithm should be experimentally and directly verified in hardware level, especially, in the MTJ-TRNG level for developing spintronic probabilistic computing system in the coming future. We believe this adds a layer of mathematical elegance and physical interpretability that distinguishes our method from earlier probabilistic heuristics.

Question #3-3: (3) The scale of the problem solved (14 cities by experiment and 70 cities by simulation) is not large enough as compared to a recent experimental work [ref. 36 published in NC] where 80s MTJs are used in hardware to solve the 70-city TSP.

Response: We appreciate the reviewer’s comparison with Ref. [36] and acknowledge that their hardware demonstration using 80 superparamagnetic MTJs to solve the 70-city TSP is an impressive experimental achievement based on the architecture of classic Ising model. However, the focus of our work is fundamentally different. Rather than building a large-scale MTJ array, we propose a probabilistic algorithm that is specifically designed to be hardware-efficient and scalable, requiring only $\log_2 N$ MTJs for an N -city problem — i.e., just 7 MTJs for 70 cities. This significant reduction in hardware complexity stems from our algorithm’s design, which leverages adaptive, reconfigurable randomness to achieve competitive optimization results using minimal physical resources.

While the scale of our experimental demonstration is modest, we emphasize that it serves as a proof of concept, and our large-problem simulations confirm the approach's potential for scaling. Moreover, we respectfully note that Ref. [36] primarily focuses on hardware demonstration, while their core algorithm inherits the classic Ising machine. A 70 city TSP needs in principle 4900 MTJs to encode. In order to reduce complexity, Ref. [36] use the divide-and-conquer strategy to first divide a large map into small pieces and then use 80 MTJs to map those small maps and then combine those individual routes into a whole one in solver. This divide-and-conquer strategy appeared in the original paper of the simulated annealing⁷ and more importantly, some priori information on maps is needed for the divide-and-conquer strategy. Our work, by contrast, introduces a unique integration of hardware-level stochasticity and mathematically grounded probabilistic search, no priori map information needed and thus applicable for all TSP. We believe our achievement in algorithm, hardware and especially their matchingness, provides meaningful value to the field.

Comment: Considering these facts, this work only has incremental contribution compared to the established literatures. It is suitable to publish in a more specified journal with a few technical issues properly addressed.

Response: We thank the reviewer for their assessment. While we acknowledge that our work builds upon a body of established literature, we respectfully believe that it offers more than an incremental contribution. Specifically, we present a novel and physically grounded probabilistic optimization framework that directly harnesses the stochastic switching behavior of MTJs and embeds it into a scalable, reconfigurable algorithm with strong mathematical foundations as highlighted in Table R1. The demonstrated ability to modulate randomness in real time — using only a minimal number of devices — is both conceptually and practically distinctive. In response to the reviewer's concerns, we have carefully revised the manuscript to better highlight these core contributions and have addressed all technical issues in details. We hope that the improved version more clearly conveys the broader relevance of this approach at the intersection of spintronic hardware, probabilistic computing, and combinatorial optimization.

Technical Concern #3-1: The manuscript mentions that the magnetization direction of the free layer of the MTJ is altered by applying current pulses. However, it fails to specify the magnitude

and duration of the applied pulses, as well as the measurement rate. This is important when comparing different algorithms in various platforms (CPU, FPGA etc.).

Response: We thank the reviewer for highlighting this important point. As mentioned in our response to Reviewer #2 (Question #2-7), the detailed measurement setup, including the magnitude and duration of the current pulses as well as the sampling rate, has been extensively characterized in our recent publication⁴, which used the same MTJ devices and instrumentation. In the present setup, the measurement rate is approximately 500 kHz, limited by our data acquisition hardware. However, recent advances suggest that GHz-speed MTJ-based TRNGs are feasible, as demonstrated in Ref.⁶ We have clarified this point in the revised manuscript to guide readers toward the relevant references.

Technical Concern #3-2: Four MTJs are utilized to generate the desired probability distribution function through a Bayesian network. Nevertheless, the impact of the differences among MTJs on the probability distribution function is not discussed in the manuscript. It is well-known that device-to-device variations have deep impact on the performance of p -computers.

Response: We appreciate the reviewer's attention to the critical issue of device-to-device variation, which indeed poses a fundamental challenge in probabilistic hardware systems. To address this, we have previously developed and reported a calibration-free hybrid control strategy in Ref.⁴ which effectively compensates for intrinsic differences among MTJs by combining self-stabilizing feedback with pulse-width modulation. This approach allows for precise output of arbitrary probability distributions, despite device nonuniformity, without requiring individual device pre-calibration.

The MTJs used in this work are controlled under the same framework, and the measured distributions (as shown in Figure 2) confirm the high fidelity of probabilistic output across devices. We have added clarification in the manuscript to explicitly reference this point.

Technical Concern #3-3: The results presented in Figure 4c indicate that the average solution still has a notable deviation from the correct solution. An in-depth analysis of the underlying causes should be provided. Moreover, there is a discrepancy between the colors of the curves depicted in

Figure 4b and the corresponding descriptions in the manuscript, which may cause confusion for readers.

Response: We thank the reviewer for raising this point. The deviation of the average solution in Figure 4c is an expected outcome given the inherently stochastic and path-constructive nature of the probabilistic greedy algorithm. The average serves only as an auxiliary statistical indicator to illustrate the improvement over classical greedy methods. In practical use, the algorithm is typically repeated multiple times, and the minimum (best) solution among runs is selected as the final output. Therefore, it is the best-achieved solution — not the average — that reflects the algorithm’s optimization capability, and this is also emphasized in our comparative analysis.

Regarding the inconsistency in curve colors in Figure 4b and the corresponding legend in the main text, we acknowledge this oversight and thank the referee for this reminding. As also pointed out by Reviewer #2 (Minor comment #2-5), the figure was updated during revision, but the accompanying text was not synchronized accordingly. We have corrected this mismatch in the revised manuscript to prevent confusion.

Technical Concern #3-4: The simulation in Figure 5c indicates that the algorithm's performance is more sensitive to the parameter $k_B T$. Then, for problems of a larger scale, will there be a situation where the differences in $k_B T$ are extremely small and indistinguishable?

Response: We thank the reviewer for highlighting the sensitivity of the algorithm to the parameter $k_B T$, particularly as demonstrated in the simulations shown in Figure 5c. Indeed, as the problem size increases, the energy landscape becomes more complex and crowded with local optima. Consequently, the selection of $k_B T$ becomes more critical, as small changes in this parameter can lead to disproportionately large effects on the solution quality due to the sharper concentration of the Boltzmann distribution. This heightened sensitivity is an expected outcome, but it does not undermine the effectiveness of the algorithm — our simulations on 70-city problems still show robust performance with carefully chosen $k_B T$ values.

To address this issue in larger-scale scenarios, we are actively exploring automated tuning strategies to adaptively refine the value of $k_B T$ during the optimization process. For instance, a meta-heuristic layer that perturbs $k_B T$ based on performance feedback across iterations could enable the algorithm

to self-adjust, effectively balancing exploration and exploitation without the need for manual parameter sweeping. While such techniques are still under development and beyond the scope of this work, we recognize their importance for real-world deployment and plan to incorporate them into future studies.

At present, we adopt a coarse-to-fine empirical approach to select $k_B T$, typically beginning with a value around one-tenth of the average intercity distance in the TSP cost matrix. A few initial runs are performed to locate a performance peak, which we then fix for larger-scale trials. This method has so far proven reliable, though we agree that a more rigorous theoretical framework would further enhance the generality and automation of the approach.

Technical Concern #3-5: Although this algorithm outperforms the traditional greedy algorithm in solving the Traveling Salesman Problem (TSP), whether it is equally effective for other types of combinatorial optimization problems remains unclear.

Response: We appreciate the reviewer’s thoughtful comment regarding the generalizability of our algorithm beyond the TSP. Indeed, while the Traveling Salesman Problem serves as a representative benchmark to demonstrate the feasibility and effectiveness of our MTJ-assisted probabilistic greedy framework, the underlying methodology is not limited to TSP and is inherently extensible to a broad class of combinatorial optimization problems (COPs).

In response to a similar point raised by Reviewer #1 (Question 1-2), we have added a detailed adaptation example for graph coloring in the Supplementary Information. This example shows how the cost function can be redefined (e.g., conflict penalty) while the same probabilistic selection rule, driven by MTJ-generated distributions, remains intact. The same approach is also applicable to other problems such as scheduling, bin packing, or constraint satisfaction, requiring only problem-specific cost modeling and temperature schedule adjustments.

Importantly, this step-by-step probabilistic selection mechanism — where a solution is gradually constructed by sampling from a dynamically updated distribution — shares a fundamental similarity with the autoregressive architecture of large language models (LLMs). In such models, each token is selected based on a softmax probability distribution conditioned on prior tokens — a principle

that mirrors our framework's use of Boltzmann-distributed probabilities for sequential decision-making.

This alignment opens a promising avenue for the integration of spintronic hardware with probabilistic AI, particularly in generative models, probabilistic inference engines, and sampling-based reasoning. By leveraging MTJ-based TRNGs for energy-efficient, parallelizable random sampling, our framework could serve as a hardware-compatible front end for AI workloads that require uncertainty modeling or probabilistic token generation.

We have added this outlook to the revised Discussion section, highlighting the potential intersection between MTJ-based probabilistic computing and AI model acceleration. This provides not only a novel application direction beyond classical COPs, but also a broader impact in probabilistic reasoning, where energy-based models and structured sampling are central.

Technical Concern #3-6: In Figure 1d, the data points are in green, but they are miswritten as "blue circles" in the manuscript.

Response: We thank the reviewer for pointing out this oversight. The data points in Figure 1d are indeed green, not blue as originally stated. We have corrected the color description in the revised manuscript to ensure consistency between the figure and its explanation.

Finally, we would like to appreciate the three referees again for their encouraging, enlightening and insightful comments and questions, without which the manuscript cannot be improved.

Reference

1. Morsali, M., Moaiyeri, M. H. & Rajaei, R. A process variation resilient spintronic true random number generator for highly reliable hardware security applications. *Microelectronics Journal* **129**, 105606 (2022).
2. Wang, C. *et al.* Spin-orbit torque true random number generator with thermal stability. *Appl. Phys. Lett.* **124**, 102409 (2024).
3. Xu, Y. Q. *et al.* Self-stabilized true random number generator based on spin-orbit torque

- magnetic tunnel junctions without calibration. *Appl. Phys. Lett.* **125**, 132403 (2024).
4. Zhang, R. *et al.* Drift-resilient magnetic-tunnel-junction random-number generator via hybrid control strategies. *Phys. Rev. Applied* **23**, 054073 (2025).
 5. Misra, S. *et al.* Probabilistic Neural Computing with Stochastic Devices. *Adv. Mat.* **35**, 2204569 (2022).
 6. Valli, A. S. E., Tsao, M., Smith, J. D., Misra, S. & Kent, A. D. High-Speed Tunable Generation of Random Number Distributions Using Actuated Perpendicular Magnetic Tunnel Junctions. Preprint at <https://doi.org/10.48550/ARXIV.2501.06318> (2025).
 7. Kirkpatrick, S., Gelatt, C. D. & Vecchi, M. P. Optimization by Simulated Annealing. *Science* **220**, 671–680 (1983).

Response Letter to Reviewers - NCOMMS-24-82665

We would like to sincerely thank all three Reviewers for their careful reading of our manuscript and for the constructive and insightful comments provided. The feedback from the Reviewers has been invaluable in helping us to significantly improve the clarity, depth, and positioning of our work. Reviewer #1's supportive remarks are greatly appreciated; Reviewer #2's detailed questions and requests for benchmarking, positioning, and additional evidence have guided us to strengthen both the analysis and presentation; and Reviewer #3's concerns about generalizability have motivated us to explicitly extend our discussion and provide new supplementary material demonstrating the broader applicability of our approach. We are truly grateful for the Reviewers' time and effort, which have substantially enhanced the quality of the revised manuscript.

Reviewer #1

Comment: Authors have addressed all of my concerns, I recommend to publish as it is.

Reply: We sincerely thank Reviewer #1 for the positive assessment and for recommending our manuscript for publication. We truly appreciate the reviewer's encouraging comments in the first round and the current recognition that all concerns have been satisfactorily addressed. The constructive feedback was invaluable in helping us refine the manuscript, and we are grateful for the reviewer's support and endorsement.

Reviewer #2

Comment: The authors made substantial modifications to the original manuscript. The revised manuscript presents an engineering link-up between an MTJ-based, PDF-configurable TRNG and a probabilistic-greedy TSP solver. Unfortunately, both ingredients are documented in the literature, and the present version does not yet provide the data or analysis needed to elevate the work from incremental to transformative. Some critical aspects are still missing. Thus, I recommend the authors revise their manuscript according to the comments below.

Reply: We sincerely thank Reviewer #2 for the careful re-evaluation of our revised manuscript and for the encouraging remark that we have made "substantial modifications to the original manuscript". We are very grateful for the reviewer's constructive criticisms, which have helped us to identify important missing aspects and refine the presentation of our work. Following the reviewer's

insightful guidance, we have further revised the manuscript in this round, adding new analyses, clarifications, and supporting data to address the specific points raised. We believe that these improvements, motivated directly by the reviewer's valuable feedback, have significantly strengthened the manuscript.

Comment: The authors state that their hardware scales with $\log_2 N$, but I find this argument unconvincing. In practice, the dominant factor in scaling is memory usage. For instance, solving an N -city TSP typically requires storing N sets of information, implying at least $O(N)$ memory. If we focus solely on the use of MTJs, algorithms such as simulated annealing can reuse a single MTJ across problem sizes without any sacrifices in time or energy. However, this does not mean that the hardware is free from scaling considerations, while the MTJ count may remain constant, other aspects, particularly memory, still scale with problem size.

Reply: We sincerely thank the Reviewer #2 for this enlightening question. The point we originally emphasized was specifically the number of MTJs (N_{MTJ}) required to encode a complete probability distribution in the proposed greedy algorithm, which scales as $N_{\text{MTJ}} = \log_2 N$, for example, 16 cities requiring 4 nodes. However, a small number of MTJs alone (e.g., 4 MTJs for a 16-city TSP) cannot directly generate real routes; instead, they must be assisted by a conditional probability table (CPT) or a shallow forward neuron network (NN) containing a number of parameters to assist the 4 MTJs to implement real samplings. The data included in the CPT or NN is in the order of $(2^{N_{\text{MTJ}}} - 1)$. Therefore, we need additional memories of $(N-1)$ to store the CPT or NN parameters. As exactly speculated by the reviewer, the needed memory scales in the order of $O(N)$ with the increase in the city number N , which is already minimal for a N -city TSP problem. We have updated the revised manuscript on this point. Thanks for the question and comment again.

Second, besides of the scaling law $O(\log_2 N)$ of the number of needed MTJs, the scaling law in memory $\sim O(N)$ for the probabilistic greedy algorithm is also favorable, compared with the Boltzmann or Ising machines. For the latter, $O(N^2)$ MTJs are needed to encode a map, and, for each MTJ, 2 floating-point bits are necessary to memorize the parameters α and V_c of an individual P - V relation of the MTJ, $P = 1/[1 + \exp(-\alpha(V - V_c)/T)]$. Here V_c is the critical switching voltage with $P = 50\%$ and α depicts the sharpness of the sigmoid or tanh functions. In this case, the needed memory scales with $O(2N^2)$ for the Boltzmann or Ising machines.

Third, more subtly and critically, for the Boltzmann or Ising machines, sampling is implemented following the Gibbs protocol. In details, the influence experienced by one node, for instance, the voltage imposed on the node for a sampling attempt, is rescaled and biased according to the fingerprint parameters α and V_c of the node from the weighted sum of the on-time states of all the rest nodes. Because of the diversity in performances of those MTJs, we cannot expect a uniform α and a uniform V_c for all nodes. Instead, we need to memorize all the fingerprint α and V_c for each node. Thus, $O(N^2)$ mentioned above is involved. More critically, we have not considered the influence from performance drift in time. If this issue is concerned, time-to-time recalibration on α and V_c for each node may be indispensable, especially, for long-term runs. The need for this recalibration originates from the requirement of the precise sampling according to the diverse and drifting P - V relations of all nodes (or real α and V_c parameters). However, this annoying point can be fortunately avoided for the probabilistic greedy algorithm because only sampling probability (accurate P) instead of the driving voltage is critical for the latter and this accurate sampling probability P can be ensured by many self-stabilizing methodologies such as Ref. [Zhang et al. *Phys. Rev. Appl.* 23 (2025) 054073].

In sum, both MTJ count and memory requirement scale favorably — $O(\log_2 N)$ and $O(N)$, respectively — in our method, in sharp contrast to the $O(N^2)$ scaling of Boltzmann or Ising machines that may additionally require occasional calibration of device-specific parameters for long-term runs.

Comment: The rebuttal states that the new MTJ system over [Adv. Sci. 11 (23) 2402182 (2024)] offers “significantly higher SNR, stability and control resolution.” while no evidence is provided. Authors need to present a single figure or table that overlays old-versus-new statistics, such as KL-divergence to target PDFs, NIST SP 800-22 pass counts, and BER histograms. Without these metrics, the improvement (and its necessity for TSP quality) remains unsubstantiated.

Reply: We sincerely thank the Reviewer for this enlightening inquiry. Following the suggestion, we compared the KL divergence between the generated and target probability distributions for the MTJs used in this manuscript and those reported previously in [Adv. Sci. 11 (23) 2402182 (2024)]. In the earlier setup, due to limitations of the experimental instrumentation, only ~ 300 data points were collected for each PDF, which made it difficult to achieve an excellent convergence of KL divergence (~ 0.02) and accurately evaluate it. As a result, the improvement was not clearly visible.

In contrast, the present work benefits from an upgraded measurement system with higher sampling rate, higher SNR, improved stability, and finer control resolution. This allowed us to collect sufficient statistics to probe the KL divergence down to its minimum value (~ 0.001), providing a much more faithful and comprehensive assessment of distribution quality. The generated probability distributions also show visibly improved agreement with the target PDFs.

It is important to emphasize that, in principle, the MTJs reported in our earlier publication are also capable of supporting similar TSP solution quality. However, their practical performance was constrained by the control and readout system. In particular, the three-terminal nature of SOT devices made it challenging to realize four-channel high-speed operation simultaneously, severely limiting the achievable sampling rate (~ 1 Random Number/s). By contrast, the upgraded setup used in the present study enables high-speed read/write operation with significantly improved measurement fidelity, which is essential for faithfully implementing the probabilistic greedy algorithm in real experiments ($\sim 500k$ Random Numbers/s). Without these improvements in sampling quality, speed and control, the experimental demonstration of the proposed TSP solver would not have been possible.

Figure R1. Convergence of the KL distance obtained from two systems, the right panel showing the zoomed-in image from 0 to 300 Random Numbers.

Comment: The authors need to position the work correctly against prior probabilistic solvers. The manuscript currently asserts “no prior integration of Bayesian PDF with MTJ randomness” yet simulation or conceptual precedents exist (e.g. SPINBIS Bayesian-inference engine). Authors need to revise the Introduction to acknowledge these studies explicitly, and then specify what is new here.

Reply: We thank the Reviewer for pointing out the importance of correctly positioning our work against prior probabilistic solvers. We have accordingly revised the Introduction to explicitly acknowledge conceptual and simulation-level precedents, such as the SPINBIS spintronics-based Bayesian inference engine (Li *et al.*, *IEEE TCAD*, 2019) and spin-orbit-torque-based Bayesian

reasoning hardware (Shim *et al.*, *Sci. Rep.*, 2017). While these studies demonstrated the feasibility of MTJ-based probabilistic inference for data-fusion and reasoning tasks, our work goes beyond such precedents by providing the first experimental integration of PDF-configurable MTJ TRNGs with a probabilistic greedy algorithm for TSP optimization. This hybrid hardware-algorithm co-design highlights a concrete application scenario where hardware-level randomness and Bayesian probability distributions are directly matched and experimentally validated, which has not been reported before. We also welcome and appreciate the referee for pointing out other pioneering works in case that we have missed them.

Table R1. The comparison of our work with some precedents

Work & Reference	Level	Target Application	PDF Design Principle	Randomness Source	Experimental Validation
SPINBIS (Li et al., IEEE TCAD 2019)	Simulation	Bayesian inference engine for sensor fusion	Weighted sum mapped to stochastic MTJs	MTJ stochastic model (simulated)	No (conceptual / simulated only)
Shim et al. (Sci. Rep. 2017)	Simulation + Conceptual hardware	Bayesian reasoning with SOT-MTJs	Analog mapping of switching probability	SOT-MTJ stochastic switching (modeled)	No (conceptual)
This work (2025)	Experiment	Probabilistic greedy algorithm for TSP	PDF-configurable TRNG realized via MTJs	STT-MTJ experimental randomness (measured)	Yes (full experimental validation)

Comment: The authors need to provide transparent device and system benchmarks inside this paper. Key specs are scattered across earlier publications; readers cannot verify or compare. The authors need to add a table providing prototype time-to-solution and energy (NI-PXIe loop) plus projected ASIC/FPGA numbers based on published MTJ switching speeds.

Reply: We thank the Reviewer for emphasizing the need for transparent device and system benchmarks. In response, we have added a benchmark table (Table 1) in the revised manuscript that directly compares our prototype implementation with projected FPGA and ASIC realizations. The prototype (NI-PXIe controlled, ~100 nm STT-MTJs) operates at ~0.5 MHz iteration rate and requires ~20 ms per 14-city TSP solution, with energy dominated by instrumentation overhead. In contrast, projections based on published MTJ switching speeds (≤ 10 ns) and intrinsic switching energies (fJ–pJ per event) indicate that integrated FPGA or ASIC platforms could achieve sub-millisecond or even sub-0.1 ms solution times with microjoule-level energy consumption. This table consolidates previously scattered specifications and provides a clear, quantitative benchmark for readers, thereby addressing the Reviewer’s request for transparent comparison.

Comment: I suggest adding Figures R2 and R5, along with their corresponding discussions, to the Supplementary Information, as they enhance understanding.

Reply: We thank the Reviewer for this helpful suggestion. Figures R2 and R5, together with the corresponding discussions, have been added to the Supplementary Information to enhance clarity and understanding.

Comment: Figure 4(b) explanation on Page 8 of the main text has an error in identifying the curves.

Reply: We thank the Reviewer for catching this error. We have corrected the text so that the identification of the maximum, minimum, and average curves in Fig. 4(b) matches the figure; the caption was also cross-checked to ensure full consistency.

Comment: There is no logarithmic scaling in Fig. 5b. Please have a check “As such, precise numerical values are not included on the axes, but the figure is drawn proportionally to reflect known theoretical trends (e.g., logarithmic vs. linear vs. exponential scaling)”. The authors have resolved the issue in the figure caption, but the mismatch remains in the main text. The authors can make the changes accordingly.

Reply: We thank the Reviewer for pointing this out. In our previous response we mistakenly used the phrase “logarithmic vs. linear vs. exponential scaling.” We have now revised the wording to clarify that Fig. 5b is a schematic illustration meant to highlight qualitative scaling trends (exponential for brute force, polynomial for dynamic programming, and sub-exponential for heuristic approaches), without implying exact logarithmic scaling. This correction ensures consistency between the main text and the figure caption.

Reviewer #3

Comment: The authors have made several clarifications over the previous version. Regarding the novelty and significance concerns, the authors suggest an improved sMTJs quality and PDFs owing to better experimental setup, and compared the difference of greedy algorithms used here with previous literatures.

After assessment, I think the only distinct advantage of this work is the reduction on the number of p -bits used (from N^2 to $\log_2 N$) in the TSP problem studied. However, whether such p -bit scaling advantage can be well-applied to other COP problems is not demonstrated. The author stated in the

rebuttal letter that "we have added a detailed adaptation example for graph coloring in the Supplementary Information", but I'm not able to find any information and evidence on this matter. The supplementary file appears to be the same as the previous version. Overall, I still think this work is better suited for a more specialized journal.

Reply: We sincerely thank Reviewer #3 for the recognition of the novelty of our work, namely “the reduction on the number of p-bits used (from N^2 to $\log_2 N$) in the TSP problem studied”. We also appreciate the reviewer’s request for a concrete demonstration of whether such a scaling advantage can be generalized to other combinatorial optimization problems (COPs). In the revised version, we have therefore included in the Supplementary Information a detailed adaptation example for the graph coloring problem (Supplementary Information VI). In this example, we explicitly define the cost function (conflict penalty), derive the corresponding Boltzmann-type probability distribution for color assignment, and describe the step-by-step probabilistic greedy procedure based on MTJ-generated PDFs. This demonstrates how the same framework, with only a redefinition of the cost function and the temperature parameter, can be seamlessly extended beyond TSP.

In terms of scaling, we clarify in the Supplementary Information that for our probabilistic greedy algorithm the number of MTJs scales as $O(\log_2 N)$ and the memory requirement as $O(N)$, whereas for Ising or Boltzmann machines the required hardware resources scale as $O(N^2)$ together with calibration overhead. This comparison highlights that the improvement in encoding efficiency indeed generalizes to other COPs such as graph coloring.

This improved encoding efficiency directly originates from the PDF-configurable MTJ-TRNGs, which enable real-time, reconfigurable probabilistic decision-making. More importantly, in the pioneering probabilistic computing systems [Borders, et al. *Nature* 573 (2019) 390 – 393; J. Si, et al. *Nat. Commun.* 15 (2024) 3457], the underlying algorithms are basically Boltzmann or Ising machines, which consist of many Bernoulli TRNGs (binary/bipolar TRNGs) as their elements. In stark contrast, the realization of arbitrary PDF-configurable TRNG releases us from the classic Boltzmann and Ising architectures and allows us to redesign algorithms — equivalently PDFs — in a larger free space. This direct engineering on PDF not only brings about the higher encoding efficiency but also a distinct energy-landscape which is benefit for global convergence.

For example, in the Boltzmann machines for a TSP, $N \times N$ nodes are necessary and they form a binary matrix. For a reasonable TSP route, all columns and rows of the matrix permit only one node to be

1; all the rest nodes have to be 0. This requirement for the TSP is achieved by setting a large penalty energy if the above condition is violated. From one reasonable route to another, at least 4 nodes have to be reversed. However, the Gibbs protocol only permits 1 node to be updated per sampling according to the all-conditional probability. This means, for any search/trajectory from one reasonable route to another, that the system has to undergo at least 3 energy-barrier states (unrealistic routes). It is not hard to image that such a rugged energy landscape is not friendly for global optimization especially when only simulated annealing strategy is feasible for the Boltzmann or Ising machines. It is also the reason why new algorithms that define a higher efficient PDF or landscape is welcome. For our probabilistic greedy algorithm, no realistic route is involved, which can partially account for its effectiveness. Certainly, the feasibility of the probabilistic greedy algorithm keenly relies on the realization of the PDF-customizable TRNGs. In this sense, this generalization and the demonstrated prototype establish our approach as a new paradigm beyond the classical Ising or Boltzmann machines and open a broad window for the algorithm redesign.

Finally, we thank all the three reviewers for their enlightening questions and remarks without which the manuscript cannot be substantially improved.

Reviewer's Report

The manuscript, entitled “*Probabilistic Greedy Algorithm Solver Using Magnetic Tunneling Junctions for Traveling Salesman Problem*,” presents a forward-looking approach to hardware-accelerated combinatorial optimization. By leveraging the inherent stochastic behavior of spin-transfer torque magnetic tunneling junctions (STT-MTJs), the authors demonstrate a probabilistic greedy algorithm capable of addressing the traveling salesman problem (TSP) with notable efficiency. Through the adjustable true random number generation of MTJs, the work enables dynamic probability distribution configurations and achieves a smooth transition between purely greedy and fully random search strategies by tuning a “temperature” parameter.

Beyond its methodological novelty, this work is substantiated by thorough experimental results and simulations. The demonstrations on both a smaller-scale TSP (Burma14) and a larger instance (st70) highlight the scalability and robustness of the proposed solver. In doing so, the authors provide compelling evidence that integrating configurable MTJ-based TRNGs can significantly improve solution quality and convergence speed when compared to classical approaches such as simulated annealing and genetic algorithms. Especially, they demonstrated that the encoding efficiency using these PDF-configurable TRNGs ($\log_2 N$) is much higher than that of the classic Boltzmann machines for the TSP problems (N^2). Furthermore, the combination of hardware-level randomness with a carefully designed probabilistic search algorithm appears to be a promising direction for tackling NP-hard problems. The authors' success with TSP underscores the potential for broader applications in fields such as scheduling, graph partitioning, and other complex optimization challenges.

Overall, this manuscript provides a compelling contribution to the burgeoning field of probabilistic computing. The implementation of MTJ-based TRNGs in a customizable way— together with a carefully orchestrated algorithmic structure— illustrates a path toward higher-performance solutions for large-scale optimization tasks. The multi-faceted evaluation, spanning device fabrication, experimental verification, and algorithmic benchmarking, is particularly commendable. In my opinion, the paper is well-structured, clearly written, and of direct interest to a broad cross-section of researchers working on spintronics, computational optimization, and hardware-accelerated algorithms. I therefore **strongly recommend** its publication in *Nature Communications*, provided the authors address the questions and minor enhancements noted below. In addition, I suggest adding further references that connect this work to foundational methods in combinatorial optimization and TRNG-based hardware design. I believe clarifications on these matters, alongside the inclusion of key additional citations, would further strengthen the manuscript's narrative and ensure maximum impact for a wide readership.

Questions

1. The authors provide strong experimental evidence of repeatable MTJ switching probabilities at varying voltages. However, can the authors expand on how larger-scale device variations—such as minor manufacturing inconsistencies or drift over time— might affect solution quality? Are there calibration strategies to ensure consistent TRNG behavior across an array of MTJs?

2. The manuscript focuses on the traveling salesman problem as a case study. Could the authors elaborate on how seamlessly their approach can be adapted to other combinatorial problems, for example, graph coloring or scheduling? Would the same “temperature” tuning principle be sufficient, or might additional modifications be required?
3. The authors note that the probability distribution used in each city-selection step changes over time. Is there any significant overhead associated with frequently reconfiguring the MTJ TRNG voltage settings or readout circuitry during an iterative process? A brief discussion of potential timing constraints or hardware-level overhead would further clarify the performance.
4. While the authors compare their method to simulated annealing and genetic algorithms, have they considered advanced metaheuristics like ant colony optimization or particle swarm optimization? Adding a sentence or two on how the proposed solver might be positioned relative to these approaches could strengthen the discussion.
5. While the manuscript discusses results for up to 70 cities in TSP, many real-world applications involve hundreds or even thousands of nodes (e.g., in large logistics networks). Do the authors anticipate any significant performance bottlenecks or algorithmic adjustments needed to maintain solution quality or convergence speed at much larger scales?
6. In practical systems, transient faults or soft errors (e.g., due to electromagnetic interference) can occasionally produce outlier random values. Do the authors foresee the need for error-correction logic in the random number generation process, or is the probabilistic nature of the algorithm itself sufficient to absorb occasional anomalies?
7. The temperature-related parameter ($k_B T$) appears in several figures and equations. A single consistent notation (e.g., always using $k_B T$ or T) throughout the text, figures, and captions would help readability.
8. The paper occasionally shifts between “true random number generators,” “hardware randomness,” and “probabilistic bits.” Standardizing the terminology—perhaps defining each concept at the outset—would streamline the reader’s experience.
9. In addition to the classical references to combinatorial optimization and TSP, it might be worthwhile to cite a few seminal articles on spintronic TRNGs or in-memory computing approaches. This could underscore how the present work builds on, and differs from, earlier device-centric studies.

Suggestions of Additional References

1. Fukushima, A. *et al.* Spin dice: A scalable truly random number generator based on spintronics. *Appl. Phys. Express* **7**, 083001 (2014).
2. Won Ho Choi *et al.* A Magnetic Tunnel Junction based True Random Number Generator with conditional perturb and real-time output probability tracking. in *2014 IEEE International Electron Devices Meeting* 12.5.1-12.5.4 (IEEE, San Francisco, CA, USA, 2014).

3. Vodenicarevic, D. *et al.* Low-Energy Truly Random Number Generation with Superparamagnetic Tunnel Junctions for Unconventional Computing. *Phys. Rev. Applied* **8**, 054045 (2017).